# A New Double-Layer Decentralized Consistency Algorithm for the Multi-Satellite Autonomous Mission Allocation Based on a Block-Chain

**DOI:** 10.3390/s22197387

**Published:** 2022-09-28

**Authors:** Fei Cheng, Xin Ning, Yunfeng Dong

**Affiliations:** 1School of Astronautics, Northwestern Polytechnical University, Xi’an 710072, China; 2Shanghai Institute of Satellite Engineering, Shanghai 201109, China; 3School of Astronautics, Beihang University, Beijing 100191, China

**Keywords:** autonomous mission allocation, decentralized consensus algorithm, satellite clusters, blockchain

## Abstract

The traditional on-board centralized-distributed mission negotiation architecture has poor security and reliability. It can easily give rise to the collapse of the whole system when the master node is attacked by malicious nodes. To address this issue, the decentralized consistency algorithms commonly used in the internet world are referred to in this paper. Firstly, four typical consistency algorithms suitable for the Internet and which are named RAFT, PBFT, RIPPLE and DPOS are selected and modified for a multi-satellite autonomous mission negotiation. Additionally, based on the above modified consistency algorithms, a new double-layer decentralized consistency algorithm named DDPOS is proposed. It is well known that the above four common consistency algorithms cannot have both a low resource occupation and high security. The DDPOS algorithm can integrate the advantages of four common consistency algorithms due to its freedom of choice attribute, which can enable satellite clusters to flexibly adopt different appropriate consistency algorithms and the number of decentralized network layers. The DDPOS algorithm not only greatly improves the security and reliability of the whole satellite cluster, but also effectively reduces the computing and communication resources occupation of the satellite cluster. Without the presence of a malicious node attack, the resource occupation of the DDPOS algorithm is almost the same as that of the RAFT algorithm. However, in the case of a malicious node attack, compared with the RAFT algorithm, the total computation and total bandwidth occupation of the DDPOS algorithm have decreased by 67% and 75%, respectively. Moreover, it is surprising that although the DDPOS algorithm is more complex, its code size is only about 8% more than the RAFT algorithm. Finally, the effectiveness and feasibility of the DDPOS algorithm in the on-board practical application are analyzed and verified via simulation experiments.

## 1. Introduction

In recent years, the number of satellites in orbit is increasing rapidly with the development of science and technology. By 1 January 2022, the total number of satellites in orbit around the world had reached 4852 [1]. With the rapid growth of the number of satellites, the traditional single satellite management and control mode is no longer applicable, which has been replaced by a multi-satellite cooperative management and control mode [2]. Hence, more and more micro-satellites with the characteristics of a small volume, a light weight and a low cost were used to replace the traditional large satellites in order to complete various command tasks [3,4,5,6].

However, the traditional ground centralized mission planning is difficult to meet the application requirements with the expansion of the scale of the satellite clusters, which is mainly reflected in the following two aspects. On the one hand, ground stations of most countries in the world are distributed within their territory, which makes the satellite unable to contact the ground station for a long time after each transit. Although the data relay satellites can be used to complete the data transmission between the satellites and the ground stations, its transmission bandwidth is insufficient. On the other hand, it is necessary for ground stations to collect all of the status information of each satellite before the mission planning, which brings enormous pressure on the data storage and transmission between the satellites and the ground stations [7,8,9,10].

The above problems are mainly caused by the inherent defects of centralization. The essence of ground centralized mission planning is the centralized control idea. The central node is responsible for collecting the state information of all nodes, making decisions and publishing results. This method will inevitably lead to a large information transmission and decision-making delay. At the same time, the efficiency of the whole system will be greatly reduced if the central node cannot work normally or the transmission link is disturbed. Compared with ground centralized mission planning, the data storage and transmission pressure of the satellite on the ground management and control can be greatly reduced by distributed mission planning, which makes up for the current defects of ground centralized mission planning. Yutao Chen et al. [11] proposed a hierarchical and distributed task-planning framework for SSA systems, and developed a customized discrete particle swarm optimization (DPSO) algorithm based on this framework. The simulation results verify its effectiveness in large-scale task planning and replanning. Du Bin et al. [12] proposed a new multi-dimensional and multi-agent cluster collaboration model (MDMA-CCM) to overcome the issues of an inflexible interactive mode, a low negotiation efficiency and a poor dynamic response capability of the satellite clusters, which could effectively increase the observation benefit and reduce the impact of task conflicts. Song, Yan-jie et al. [13] proposed a general data-driven framework-imaging satellite mission planning framework (ISMPF) for solving imaging mission planning problems, and designed test examples for the measurement and control, and data downlink missions in order to verify the validity of ISMPF. Dalin Li et al. [14] proposed a multiagent deep reinforcement learning (MADRL)-based method to solve the problem of scheduling the real-time multi-satellite cooperative observation, which could greatly reduce the communication overhead and achieve better use of satellite resources.

It is well known that the mission planning for satellite clusters is a complex NP-hard problem. In most studies on distributed mission planning, new or improved multi-objective optimization algorithms have been emphasized, with attention given to improving the overall observation benefits while ignoring the uncertainty of satellite mission planning caused by the complex and changeable space environment [15]. The impact of particle radiation on satellites is very serious. This radiation, including protons and electrons, will interfere with the communication data, which is likely to produce wrong negotiation data during the mission negotiation process [16]. At the same time, the satellite communication may also be maliciously attacked or tampered with at any time during the mission negotiation process, resulting in an inconsistent system communication, which is primarily contributed to the collapse of the whole satellite cluster communication system [17]. The traditional distributed mission planning is also difficult to solve the above problems, so the decentralized consistency algorithm is needed to solve the problems.

The United States put forward the concept of the “resiliency space system” as early as 2012, and issued the white paper “Resiliency and Disaggregated Space Architectures” in August 2013, in which the core idea is to use small satellite clusters with scattered functions to replace the traditional single large satellite [18,19]. The idea of decentralization is one of the core technologies supporting the transformation.

As the decentralized system belongs to a distributed system, it also meets the CAP theorem (three properties: consistency, availability and partition tolerance) proposed by Eric Brewer in 2000 [20]. We can learn from the CAP theorem that consistency, availability and partition tolerance cannot be satisfied simultaneously. Hence, one of three properties must be weakened in practical engineering. We refer to the POW algorithm applied in Bitcoin when applying the idea of decentralization to on-board autonomous mission planning. Consistency is chosen to weaken [21], which means that the satellites in the cluster do not need to be always consistent, but only need to reach an agreement at set intervals.

The basic principle of the POW algorithm can be understood as that all of the nodes start solving a mathematical problem at the same time [22,23]. The success rate of the problem solving is directly proportional to the computational power of the nodes. The first solved node will publish a block, and the block content will be recorded after being verified by most members, so as to reach a consensus. The POW algorithm has the following two great advantages. One is security. As the success rate of the problem solving is directly proportional to the computing power, it means that the total computing power must be occupied by at least 51% if the whole system floor is needed to be controlled. As we all know, it is almost impossible for a single individual to occupy more than half of the total computing power in the world. Moreover, the cost of occupying 51% of the total computing power in the world is much higher than the benefit of controlling the whole system floor. Hence, the POW algorithm is very safe for the whole system. The second is highly decentralized. The voice of the system changes constantly, which is closely related to the speed of the problem-solving at each node. Hence, the system does not depend on any node and is highly decentralized.

However, efficiency is a fatal weakness of the POW algorithm. A large number of hash operations are used by the nodes in the problem-solving process, which are completely irrelevant to the recorded block content, so these computing resources for the hash operations are wasted. In fact, the computing resources of each satellite is very rare, so the POW algorithm is not selected for on-board autonomous mission planning in this paper. Hence, it is necessary to try other commonly used decentralization consistency algorithms.

Commonly used decentralization consistency algorithms mainly include RAFT [24,25,26,27,28], PBFT [29,30,31,32,33], RIPPLE [34,35,36,37,38] and DPOS [39,40,41,42,43]. All of the above four algorithms have good results in ensuring data consistency. However, there are still the following problems in applying these decentralization consistency algorithms to on-board mission planning. On the one hand, all of the above four decentralized consistency algorithms are designed to solve the disadvantages caused by internet centralization in the early stage. The communication link between satellites may be affected by many external conditions, including the periodic orbit motion of the satellite, antenna pointing, etc., which is quite different from the communication links between the nodes on the Internet. Hence, these four decentralization consistency algorithms cannot be transplanted directly to the satellite for mission planning. On the other hand, both the inter-satellite communication resources and on-board computing resources are very limited. The algorithm structure of RAFT is very simple. In addition, the communication and computing resource occupation of the RAFT algorithm is generally less than the other three algorithms. However, the RAFT algorithm cannot have the ability to tolerate the Byzantine fault, as only fault-tolerant fault nodes are supported and fault-tolerant malicious nodes are not supported, which result in a significant reduction in the security and reliability of information [29]. The other three algorithms can tolerance the Byzantine fault, which have a strong resistance to the attacks of the faulty nodes and malicious nodes. However, compared with the RAFT algorithm, these three algorithm structures are much more complex and the resource occupation of each is generally quite larger. Hence, we aim to find a new decentralized consistency algorithm that combines the advantages of the above four algorithms. The new algorithm has the characteristics of safety, reliability and a low resource occupation, which can be well applied to on-board autonomous mission planning.

In this paper, we mainly looked at the following two aspects. First, the above four common decentralized consistency algorithms including RAFT, PBFT, RIPPLE and DPOS were modified according to the actual inter-satellite communication requirements, so that they could be applied to on-board autonomous mission planning. We considered the main characteristics of each algorithm and ignore the unimportant characteristics of each algorithm during the modification process. Second, we proposed a new double-layer decentralized consistency algorithm which we named DDPOS based on the four modified decentralized consistency algorithms in the first step. The DDPOS algorithm had a new organizational structure, which was different from the above four decentralized consistency algorithms. It should be noted that DDPOS had a two-layer structure, and each layer could choose the most suitable one from the above four modified decentralized consistency algorithms. The DDPOS algorithm could integrate the advantages of the above four decentralized consistency algorithms.

The remaining part of the paper proceeds as follows. In Section 2, the organizational structure of DDPOS is introduced. In Section 3, the four modified consistency algorithms we selected for DDPOS (named RAFT, PBFT, RIPPLE and DPOS) are described. In Section 4, the resource occupation and resistance to malicious nodes of the new algorithm and the four traditional consistency algorithms are compared and analyzed. In Section 5, the full text is summarized, and the future research direction is put forward.

## 2. The New Double-Layer Decentralized Consistency Algorithm (DDPOS)

The requirement of decentralization is that satellites can exchange information and reach consensus by using the inter-satellite communication link. However, the state of the communication link between satellites changes from time to time, as it is easy to be disturbed by external environmental factors. Moreover, the network topology structure among satellites is also time-varying, which makes it impossible for a satellite to communicate with all other satellites at anytime and anywhere. Hence, all satellites are divided into different groups which called the satellite clusters. Each satellite can freely form, join or exit the satellite cluster according to the mission requirements and the distance between itself and other satellites. The decentralized organization relationship is adopted within and between satellite clusters to improve the resiliency of the whole satellite system. Based on the theory of resiliency space proposed by the AFSPC, we designed the DDPOS algorithm to meet the tactical communication requirements of large-scale satellite clusters as much as possible in the case of malicious node interference [43,44]. The basic communication architecture we designed for satellite clusters is shown in Figure 1.

In this communication architecture, each satellite cluster can be composed of different numbers of heterogeneous satellites. Moreover, one or more satellites are generally selected from each satellite cluster in order to manage the whole cluster. All satellites can join or exit the cluster freely according to the actual situation. In this section, the short-range omni-directional antenna is used for satellites in the satellite cluster to communicate with each other, which is very convenient for information exchange between satellites.

It should be noted that the information exchange between satellite clusters is not necessary, as it is determined by actual requirements. In most cases, the completion of the mission requires the cooperation between multiple satellite clusters. However, in special cases, the mission can also be completed independently through only one satellite cluster. Even the cluster can be composed of one satellite. It is well known that the topological relationship between multiple satellite clusters in the same orbital plane is generally relatively stable since all satellites must revolve periodically around the earth. On the contrary, the topological relationship between satellite clusters from different orbital planes changes periodically with time, which means that sometimes the communication link between satellite clusters can be interrupted for a period. The main missions completed among satellite clusters include data relay, intelligence sharing, mission negotiation, assistance in planning, assistance in data processing, navigation and so on.

There are four traditional consistency algorithms commonly used on the Internet, which are called RAFT, PBFT, RIPPLE and DPOS, respectively. The above four consistency algorithms cannot be directly used for on-board mission planning due to the particularity of the satellite and its environment. Hence, these four consistency algorithms need to be modified to adapt to on-board mission planning, which are described in detail in Section 3. Moreover, it should be noted that the bandwidth and computing resources of a single satellite is very limited, and communication security is a key problem for onboard mission planning. These four traditional consistency algorithms have their own shortcomings. The RAFT algorithm occupies less resources but has poor security. The PBFT algorithm occupies less resources and has good security. However, the PBFT algorithm requires the leader to have a lot of computing resources, which puts great pressure on the leader. Compared with the PBFT algorithm, the resource occupation of each satellite is more balanced in the RIPPLE and DPOS algorithms. However, the total resource occupations of both the RIPPLE and DPOS algorithms are much larger than that of the PBFT algorithm. Hence, we expect to design an algorithm with a new framework which can combine the advantages of these four traditional consistency algorithms. 

Based on the above expectations, a new double-layer decentralized consistency algorithm named DDPOS is innovatively proposed, according to the above network topology structure among the satellites. In the DDPOS algorithm, each satellite cluster can flexibly adopt an appropriate consistency algorithm according to its own actual situation and needs, which constitutes the first-layer decentralized network. The appropriate consistency algorithm can be any one of the modified RAFT, the modified PBFT, the modified RIPPLE and the modified DPOS. The leaders of each satellite cluster are gathered to form a second-layer leader group, which constitutes a second-layer decentralized network. The second-layer leader group can also flexibly adopt the appropriate consistency algorithm. Moreover, the network topology structure is expandable. In theory, the leaders of each second-layer leader groups are gathered to form a third-layer leader group. The third-layer leader group can also flexibly adopt the appropriate consistency algorithm. The network topology structure of the two-layer DDPOS algorithm is shown in Figure 2. 

Main workflow of the DDPOS algorithm can be described in the following steps.

Step 1: Mission initiation process. Each satellite node finds its appropriate mission set according to the current situation. The appropriate mission set is also called the desire mission set.

Step 2: Consensus process in the satellite cluster. The appropriate mission set of each satellite is released to the leader of the satellite cluster to which it belongs. According to the desire mission set of each satellite, the leader of the satellite cluster reasonably allocates missions to all satellites by using the mission allocation strategy, and makes each satellite in the cluster reach a consensus through the selected consistency algorithm.

Step 3: Consensus process among the satellite clusters. The leaders of each satellite cluster negotiate missions on behalf of their own satellite clusters, and finally reach a consensus among satellite clusters through the selected consistency algorithm. It should be noted that this is only the second-layer negotiation architecture, which can be extended to the consensus negotiation of the third-layer, the fourth-layer and so on, in theory.

Step 4: Release of the missions. The release process is a reverse step of the initiation process. The mission is first released in the satellite cluster which belongs to the highest-layer architecture, and then go down layer by layer until the mission is successfully released to each satellite node.

The improved DDPOS algorithm makes all satellites reach a consensus through two behaviors: the mission origin behavior and the consensus behavior. The pseudo code of the above two behaviors can be shown in Algorithms 1 and 2, respectively.
**Algorithm 1.** The mission origin behavior**Input:** Satellite i **Output:** desired missions of Satellite i **1: If** Satellite i has missions to execute currently **then**
**2:   return**
**3: End If**
**4: If** there is currently no leader in the satellite cluster to which Satellite i belongs **then**
**5:   return**
**6: End If**
**7:** Satellite i calculates the observation time windows of the missions **8:** Satellite i calculates the costs and profits of missions **9:** Satellite i calculates the profit-cost ratios of missions **10:** satellite i choose missions with high profit-cost ratios and small mission execution times as its desire missions **11: If** Satellite i is the follower **then**
**12:**   Satellite i submits its desired missions to the leader of the satellite cluster to which it belongs, including observation time windows, profits and costs **13: Else If** Satellite i is the leader **then**
**14:**   Satellite i writes its desired missions into the desire mission set **15: End If**

The mission origin behavior of the modified DDPOS algorithm can be seen in Algorithm 1. It should be noted that the mission origin behavior is a periodic behavior. Each satellite needs to perform the mission origin behavior spontaneously and at regular intervals.

The pseudo code of the modified DDPOS algorithm can be shown in Algorithm 1.

The consensus behavior of the modified DDPOS algorithm can be seen in Algorithm 2. It should be noted that the selected consistency algorithm could be any one of the modified RAFT, the modified PBFT, the modified RIPPLE or the modified DPOS algorithms. The specific consensus process of the four modified consistency algorithms can be seen in Section 3.1, Section 3.2, Section 3.3 and Section 3.4.
**Algorithm 2.** The consensus behavior of the DDPOS algorithm**Input:** The set of all satellite nodes, the total number of satellite nodes N_S_, the number of satellite clusters N_SC_, the consistency algorithm selected for each satellite cluster **Output:** Consensus result //Consensus process in the satellite cluster **1: For** i = 1 to N_SC_
**2:**   Satellite cluster i reaches a consensus by using the selected consistency algorithm **3: End For**
//Consensus process among the satellite clusters **4:** The leaders of all satellite clusters reach a consensus by using the selected consistency algorithm //Release of the missions **5:** The related missions are released to the leaders of all satellite clusters **6: For** i = 1 to N_SC_
**7:**   The leader of satellite cluster i releases related missions to other satellites of the satellite cluster to which it belongs **8: End For**
**9: End while**

## 3. Four Modified Common Consistency Algorithms Selected for DDPOS

It is well known that the communication environment of the satellite is quite different from that of the Internet. More importantly, contrary to the original consistency algorithm, the modified consistency algorithm should have the ability to enable all satellite nodes to reach a consensus on the mission allocation results. Hence, in this section, we only consider the main characteristics of each algorithm and ignore other unimportant characteristics of each algorithm during the modification process. 

In summary, the modifications are mainly divided into three aspects. The first is that the mission origin process and the mission allocation process are inserted into the four original consistency algorithms. The mission origin process is responsible for calculating the profit and cost of each mission executed by each satellite. The mission allocation process is responsible for reasonably allocating the missions to each satellite based on the benefits and costs of the missions. The mission allocation results are written into the mission block. The second is that these four modified consistency algorithms all adopt the leader election mechanism of the RAFT algorithm. In order to adapt the leader election mechanism to the on-board mission planning, the relevant parameters of the leader election mechanism are modified, and this can be seen in Section 3.1 for further details. The third modification is that only the main characteristics of each of the four original consistency algorithms are considered. The modified RAFT algorithm is chosen as the basic framework in the PBFT, RIPPLE and DPOS algorithms. The main characteristic of the PBFT algorithm, which is called the three-stage consensus process, is added into the modified RAFT algorithm in order to form the modified PBFT algorithm. The main characteristic of the RIPPLE algorithm, which is called the establishment of the leader group, is added into the modified PBFT algorithm to form the modified RIPPLE algorithm. The main characteristic of the DPOS algorithm, which is called the signature verification mechanism, is added into the modified RIPPLE algorithm to form the modified DPOS algorithm.

### 3.1. The Modified RAFT Algorithm

The RAFT algorithm is a consistency algorithm suitable for both a private chain and an alliance chain, which is quite suitable for small satellite clusters. The basic principle of the RAFT algorithm is as follows. First, one leader node, which is called the leader in the cluster, will be elected. The leader is responsible for the cluster management and the mission block publishing on the condition that it can be trusted. A new trusted leader can be re-elected once the current leader’s abnormal performance is found, such as a node failure. Compared with the other three algorithms, the structure of the RAFT algorithm is simpler. The core technology of the traditional RAFT algorithm is the leader election mechanism. The RAFT algorithm is modified for the multi-satellite autonomous mission allocation, in this section. The basic workflow of the RAFT algorithm is shown in Figure 3.

Firstly, each satellite in the satellite cluster is regarded as a node to participate in the negotiation. There are three identities of nodes in the cluster which are the leader, the follower and the candidate, respectively. The leader is mainly responsible for establishing periodic contact with other followers by using the heartbeat mechanism to maintain the whole network structure. The leader organizes the mission negotiation among nodes in the cluster, and releases the mission blocks containing negotiation results to the followers after the negotiation. The follower is mainly responsible for initiating missions. The followers initiate mission requests to the leader, and then negotiate the missions with other followers. Moreover, the follower is also responsible for verifying and recording the mission blocks released by the leader. The candidate is an intermediate state between the leader and the follower. It is well known that the leader will send periodic heartbeat messages to all followers. The follower will judge that it has lost contact with the leader and originates vote if it does not receive the heartbeat message from the leader for more than a certain period of time. At the same time, the identity of the follower will be automatically transformed into a candidate. Following the election, it will be elected as a leader or continue to be the follower.

The main workflow of the RAFT algorithm can be described as the following steps (Figure 3).

Step 1: Status initialization. The identities of all satellite nodes in the satellite cluster are initialized to be the followers. Moreover, each node is set with an election timeout timer. the election timeout time of each node will be generated randomly and then reset during the status initialization. The election timeout can be understood as the amount of time that a follower waits until becoming a candidate.

Step 2: Election after timeout. Election timeout has first occurred in Satellite I, which means that Satellite I does not receive any heartbeat messages from the leader before its election timeout has occurred. In this case, Satellite I automatically changes its identity to the candidate and originates a vote request to all other nodes in the cluster.

Step 3: Vote process. Other nodes in the cluster can vote in favor or against the vote request. Satellite I is successfully elected and automatically changes its identity to the leader when more than half of the nodes in the cluster vote in favor of the vote request originated by it.

Step 4: Heartbeat maintenance. Once Satellite I becomes the leader, it will send periodic heartbeat messages to other nodes to prevent new elections to occur. Once the candidate receives the heartbeat message, it will automatically change its identity to become a follower. Once the follower receives the heartbeat message, it will reset its election timeout and reply the heartbeat response to the leader.

Only one leader can exist in the RAFT algorithm at any one time. Each node in the cluster is also given the attribute of a term number in order to prevent the simultaneous existence of no less than two leaders. The term number is a strictly increasing continuous integer value. Each new election is considered as a new term, which means that the term number needs to be increased by one. If a leader receives a message from a node with a higher term number than itself, it will consider that a new leader already exists in the current cluster, and then change its identity to become a follower. This usually occurs on the condition of network partitions. When a network partition occurs, different network partitions will conduct their own elections since they are not connected to each other. In this case, one node in each partition is elected to be the leader. There will be more than one leader in the whole cluster if the network partition is repaired. At this time, the number of leaders in the cluster is reduced to one by comparing the term numbers. In other words, a leader will automatically change its identity to become a follower if its term number is less than that of another leader.

It should be noted that, some unexpected circumstances may lead to an election failure during the election process, which means that the leader is not successfully elected after the election. This usually happens when there are two candidates in the cluster and they receive the same number of votes. The candidates will increment their term numbers by one and start a new election again, if no leader is elected successfully in the cluster after the election timeout. At the same time, the election timeout of all nodes will be reset. The time of the election timeout for each node is randomly set in order to prevent such a situation to occur, and the value range is generally 150–300 ms.

In fact, the RAFT algorithm is modified as follows, considering the shortage of network resources among the satellite clusters.

(1) Extend the sending period of heartbeat messages. The network between satellites in the satellite cluster is unstable due to the complex and changeable environment in space. If we continue to use a millisecond-level sending period of heartbeat messages adopted by the traditional RAFT algorithm in actual on-board autonomous mission negotiation, a slight network instability or response delay will be considered disconnected, which means that a large number of nodes will participate in the frequent re-election and vote process. It will lead to a large waste of satellite bandwidth and computing resources. It is well known that the bandwidth and computing resources of satellites are very limited. Hence, the sending period of the heartbeat message is extended from the millisecond-level to the minute-level, which means that heartbeat messages are sent to other followers every 60 s.

(2) Extend the time of election timeout. According to the principle of the RAFT algorithm, the re-election process is triggered when a node cannot receive the heartbeat message before its election timeout. Hence, the time of election timeout is positively correlated to the sending period of the heartbeat messages. The time of the election timeout also needs to be increased as the sending period of the heartbeat messages is increased. In this article, the time of the election timeout is set between 90 s and 180 s.

(3) Heartbeat messages without mission blocks. In the traditional RAFT algorithm, the mission blocks are contained in the heartbeat messages and sent together to other followers by the leader. The release of the mission blocks is also delayed with the extension of the sending period of the heartbeat messages, which results in a significant increase in the task response time. Hence, the release of the mission blocks and heartbeat messages are separated independently in this paper. Moreover, two release modes which are called the periodic release and the emergency release, respectively, are set for the release of the mission blocks. The periodic release means that the leader releases the mission blocks to the other followers according to the set release period. If there are no new missions or mission adjustments, an empty block will be released by the leader. The emergency release is for emergency missions, which means that a mission block will be released to other followers by the leader immediately, if it is an emergency.

The improved RAFT algorithm makes all satellites reach a consensus through two behaviors: the mission origin behavior and the leader decision behavior. The pseudo code of the above three behaviors can be shown in Algorithms 1 and 3, respectively.

The mission origin behavior of the modified RAFT algorithm is the same as that of the DDPOS algorithm, which can be seen in Algorithm 1.
**Algorithm 3.** The leader decision behavior of the modified RAFT algorithm**Input:** The set of all satellite nodes, the total number of satellite nodes N_S_
**Output:** Consensus result **1: Switch** ucStep **do**
**2:   Case** 0 **3:      If** the decision timer does not time out **then**
**4:      Break**
**5:   End If**
**6:  ** Restart the decision timer **7: If** the leader does not receive any desired missions **then**
**8:      Break**
**9:   End If**
**10:      ** ucStep = 1 **11:    Break**
**12: Case** 1 **13:   ** The leader allocates missions according to the desired mission set **14:   ** The leader writes the mission allocation results into the mission block **15:   ** The leader broadcasts the mission block to other followers through inter-satellite **16:** communication **17:   ** The leader assesses whether the missions assigned to itself can be executed **18: If** the leader can complete the missions assigned to itself **then**
**19:      ** ConfirmNumber = 1 **20:      ** RefuseNumber = 0 **21:    Else If**
**22:   ** ConfirmNumber = 0 **23:   ** RefuseNumber = 1 **24: End If**
**25:   ** ucStep = 2 **26: Break**
**27: Case** 2 **28: If** ConfirmNumber = N_S_ && RefuseNumber = 0 **then**
**29:   ** The leader releases the mission allocation results officially to all satellite nodes **30:   ** Restart the decision timer **31:   ** ucStep = 0 **32:    Break**
**33: Else If** ConfirmNumber + RefuseNumber = N_S_
**then**
**34:   ** ucStep = 1 **35:    Break**
**36:    Else If** the receive timer times out **then**
**37:      ** Restart the decision timer **38:      ** ucStep = 0 **39:   Break**
**40:   End If**
**41:   Break**
**End Switch**

The leader decision behavior of the modified RAFT algorithm can be seen in Algorithm 3. It should be noted that the leader decision behavior is also a periodic behavior, which needs to be performed in each simulation step. The initial value of the ucStep is 0. Once the leader broadcasts the mission block to other followers through inter-satellite communication (Line 15 in Algorithm 3), each follower needs to assess whether the missions assigned to itself can be executed, and then reply its assessment result to the leader. Specifically, if the follower can complete the missions assigned to itself, it will reply with a confirmation message to the leader, which will add 1 to the value of the ConfirmNumber. If not, it will send a refusal message to the leader, which will decrement the value of the RefuseNumber by 1.

### 3.2. The Modified PBFT Algorithm

The PBFT algorithm is a consistency algorithm suitable for both private chains and alliance chains. The leader node may make the wrong decisions due to the malicious transmission of wrong information by some nodes, such as in the case of the RAFT algorithm, which is also called a Byzantine problem. A Byzantine problem is a consistency preserving problem, which mainly considers the existence of malicious nodes. The Byzantine problem can be solved effectively by using the PBFT algorithm. The whole cluster can reach a consensus by using the PBFT algorithm on the condition that a few nodes transmit error information maliciously.

The core mechanism of the PBFT algorithm is a three-stage consensus process. In this paper, the related algorithms of RAFT are used to complete the underlying network maintenance in order to optimize the amount of code, which means that one leader node is also elected to manage the whole cluster in the PBFT algorithm as in the RAFT algorithm. At the same time, the three-stage consensus process is added into the PBFT algorithm. 

The main workflow of the PBFT algorithm can be described as the following steps (Figure 4).

Step 1: Initialization. The whole satellite cluster is composed of a leader and several followers, which constitutes the initial state of the PBFT algorithm. It should be noted that the leader in the PBFT algorithm is still elected through the election mechanism of the RAFT algorithm.

Step 2: Pre-release of the mission blocks. The leader releases the mission blocks to all followers in the form of a broadcast. The mission allocation results for all nodes are stored in the mission blocks.

Step 3: Verification of the block. Each follower needs to verify the mission blocks after receiving them. The verification content mainly focuses on whether the assigned missions are reasonable. Each satellite node plans the missions assigned to itself by using the mission planning algorithm and judges whether it can complete it successfully. If the satellite node can complete the missions assigned to itself, the block verification result will be recorded as 1. If not, the block verification result will be recorded as 0. Finally, each satellite node releases its block verification result to all satellite nodes except itself, in the form of broadcast. 

Step 4: Verification of the support rate. Each satellite node starts to calculate the support rate of the leader separately, after receiving the block verification results released by all of the other nodes. The calculation result of the support rate is stored in the support rate verification result of each satellite node, respectively. For each satellite node, the support rate verification result will be recorded as 1 if more than half of the block verification results received are recorded as 1. In other words, it shows that the mission allocation results released by the leader are reasonable and recognized by more than half of the satellite nodes. If not, the support rate verification result will be recorded as 0, which means that the mission allocation results released by the leader are not recognized by most satellite nodes in the cluster. Finally, each satellite node releases its support rate verification result to all satellite nodes except itself, in the form of a broadcast.

Step 5: Verification of the support result. Each satellite node starts to verify the support result separately, after receiving the support rate verification results released by all of the other nodes. For each satellite node, the verification result of the support result will be recorded as 1 if more than half of the support rate verification results received are recorded as 1. In other words, it shows that the leader is supported by most nodes in the cluster. If not, the verification result of support result will be recorded as 0, which means that the leader is not supported by most nodes in the cluster. Finally, each satellite node releases its verification result of the support result to the leader.

Step 6: Release and execution of the missions. The leader receives and analyzes the verification result of the support result released by all of the other followers. It is considered that the current mission allocation result released by the leader will be successfully validated if more than half of the verification results of the support results are recorded as 1. In this case, the leader will officially release the mission allocation result, and then each satellite node will execute the mission according to the mission allocation results. It is considered that the verification of the current mission allocation result released by the leader will fail if more than half of the verification results of the support results are recorded as 0. In this case, the missions allocated by the leader will be not executed and a new leader will be elected in the satellite cluster.

It should be noted that the leader can be replaced in the PBFT algorithm. Whether the leader should be replaced depends on the degree of support for the leader from other followers. The support rate of the leader is calculated by each satellite node, respectively, during the process of the support rate verification. If the support rate for the leader is low, it indicates that the mission allocation strategy of the leader is problematic, the cluster will automatically translate its identity from the leader to the follower. Once the current leader is revoked, the RAFT election mechanism will be used to elect a new leader in the cluster (See Section 3.1 for details).

Hence, the whole system is prevented from crashing because the leader’s error can be effectively prevented by using the three-stage consensus mechanism. Once a leader fails, a new leader can be elected through the leader election mechanism, which will effectively improve the security of the overall system.

The improved PBFT algorithm makes all satellites reach a consensus through three behaviors: the mission origin behavior, the leader decision behavior and the follower behavior. The pseudo code of the above three behaviors can be shown in Algorithms 1 and 4–7, respectively. The follower behavior can be divided into three parts: the follower block verification behavior (Algorithm 5), the follower support rate verification behavior (Algorithm 6) and the follower support result verification behavior (Algorithm 7).

The mission origin behavior of the modified PBFT algorithm is the same as that of the DDPOS algorithm, which can be seen in Algorithm 1.
**Algorithm 4.** The leader decision behavior of the modified PBFT algorithm**Input:** The set of all satellite nodes, the total number of satellite nodes N_S_
**Output:** Consensus result **1: Switch** ucStep **do**
**2: Case** 0 **3:   If** the decision timer does not time out **then**
**4:      Break**
**5:   End If**
**6:** Restart the decision timer **7:   If** the leader does not receive any desired missions **then**
**8:      Break**
**9:   End If**
**10:      ** ucStep = 1 **11:    Break**
**12: Case** 1 **13:** The leader allocates missions according to the desired mission set **14:** The leader writes the mission allocation results into the mission block // Verification of the block **15:** The leader broadcasts the mission block to other followers through inter-satellite **16:** communication **17:** The leader assesses whether the missions assigned to itself can be executed **18:    If** the leader can complete the missions assigned to itself **then**
**19:      ** PrepareConfirmNumber = 1 **20:      ** PrepareRefuseNumber = 0 **21:    Else If**
**22:      ** PrepareConfirmNumber = 0 **23:      ** PrepareRefuseNumber = 1 **24:    End If**
**25:      ** ucStep = 2 **26:    Break**
**Case** 2 **27: //** Verification of the support rate **28:    If** PrepareConfirmNumber * 2 > N_S_ || The receive timer times out **then**
**29:    If** PrepareConfirmNumber > PrepareRefuseNumber **then**
**30:      ** The support rate verification result of the leader is recorded as 1 **31:      ** CommitConfirmNumber = 1 **32:      ** CommitRefuseNumber = 0 **33:    Else If**
**34:      ** The support rate verification result of the leader is recorded as 0 **35:      ** CommitConfirmNumber = 0 **36:      ** CommitRefuseNumber = 1 **37:    End If**
          The leader broadcasts its support rate verification result to other satellite nodes **38:      ** through inter-satellite **39:      ** communication **40:      ** PrepareConfirmNumber = 0 **41:      ** PrepareRefuseNumber = 0 **42:      ** Restart the receive timer **43:      ** ucStep = 3 **44:    End If**
       **Break**
**45: Case** 3 **46**:// Verification of the support result **47:    If** CommitConfirmNumber * 2 > N_S_ || The receive timer times out **then**
**48:    If** CommitConfirmNumber > CommitRefuseNumber **then**
**49:      ** The verification result of support result of the leader is recorded as 1 **50:      ** FinalConfirmNumber = 1 **51:      ** FinalRefuseNumber = 0 **52:    Else If**
**53:      ** The verification result of the support result of the leader is recorded as 0 **54:      ** FinalConfirmNumber = 0 **55:      ** FinalRefuseNumber = 1 **56:    End If**
**57:      ** CommitConfirmNumber = 0 **58:      ** CommitRefuseNumber = 0 **59:      ** Restart the receive timer **60:      ** ucStep = 4 **61:    End If**
**62:    Break**
**63: Case** 4 **64:    If** FinalConfirmNumber * 2 > N_S_
**then**
**65:    If** FinalConfirmNumber > FinalRefuseNumber **then**
**66:      ** The leader releases the mission allocation results officially to all satellite nodes        **Else If**
**67:      ** The mission allocation fails, and a new leader will be selected through the **68:      ** leader election mechanism of the RAFT algorithm **69:    End If**
**70:      ** FinalConfirmNumber = 0 **71:      ** FinalRefuseNumber = 0 **72:      ** Restart the receive timer **73:      ** ucStep = 0 **74:    End If**
          **If** The receive timer times out **than**
**75:         ** The mission allocation fails, and a new leader will be selected through the             leader **76:         ** election mechanism of the RAFT algorithm **77:         ** FinalConfirmNumber = 0 **78:         ** FinalRefuseNumber = 0 **79:         ** Restart the receive timer **80:         ** ucStep = 0 **81:    Break**
**82:    End If**
       **Break**
**End Switch**

The leader decision behavior of the modified PBFT algorithm can be seen in Algorithm 5. It should be noted that the leader decision behavior is also a periodic behavior, which needs to be performed in each simulation step. The initial value of the ucStep is 0.
**Algorithm 5.** The follower block verification behavior of the modified PBFT algorithm**Input:** Follower i **Output:** Confirm or refuse message **1:** Follower i assesses whether the missions assigned to itself can be executed **2: If** Follower i can complete the missions assigned to itself **then**
**3:** Follower i sends a confirmation message to all other satellite nodes **4: Else If**
**5:** Follower i sends a refusal message to all other satellite nodes **6: End If**

The follower block verification behavior of the modified PBFT algorithm can be seen in Algorithm 5. Once the leader broadcasts the mission block to other followers through inter-satellite communication (Line 15 in Algorithm 4), each follower need to assess whether the missions assigned to itself can be executed, and then reply its assessment result to all other satellite nodes. Specifically, if the follower can complete the missions assigned to itself, it will reply a confirmation message to all other satellite nodes, which will add 1 to the value of the PrepareConfirmNumber of each satellite node. If not, it will reply a refusal message to the leader, which will decrement the value of the PrepareRefuseNumber of each satellite node by 1.
**Algorithm 6.** The follower support rate verification behavior of the modified PBFT algorithm**Input:** Follower i **Output:** Confirm or refuse message **1: If** PrepareConfirmNumber * 2 > N_S_ || The receive timer times out **then**
**2:   If** PrepareConfirmNumber > PrepareRefuseNumber **then**
**3:     ** The support rate verification result of Follower i is recorded as 1 **4:   Else If**
**5:     ** The support rate verification result of Follower i is recorded as 0 **6:   End If**
**7:     ** Follower i broadcasts its support rate verification result to all other sat **8:     ** ellite nodes **9:     ** Restart the receive timer        **End If**

The follower support rate verification behavior of the modified PBFT algorithm can be seen in Algorithm 6. It should be noted that not only the leader needs to verify the support rate (Line 27–44 in Algorithm 4), but also all followers need to verify the support rate. The follower support rate verification behavior is essentially the same as the verification of the support rate of the leader ‘s decision behavior (Line 27–44 in Algorithm 4). Each follower receives the confirmation (Line 3 in Algorithm 5) or refusal message (Line 5 in Algorithm 5) from other satellite nodes, and then make statistics. Specifically, if the number of the confirmation messages is larger than that of the refusal messages, it will reply 1 to all other satellite nodes, which will add 1 to the value of the CommitConfirmNumber of each satellite node. If not, it will reply 0 to all other satellite nodes, which will decrement the value of the CommitRefuseNumber of each satellite node by 1.
**Algorithm 7.** The follower support result verification behavior of the modified PBFT algorithm**Input:** Follower i **Output:** Confirm or refuse message **1: If** CommitConfirmNumber * 2 > N_S_ || The receive timer times out **then**
**2:   If** CommitConfirmNumber > CommitRefuseNumber **then**
**3:     ** The verification result of the support result of Follower i is recorded as 1 **4:   Else If**
**5:     ** The verification result of support result of Follower i is recorded as 0 **6: End If**
**7:** Follower i sends its verification result of the support result to the leader **8:** Restart the receive timer **End If**

The follower support result verification behavior of the modified PBFT algorithm can be seen in Algorithm 7. It should be noted that not only the leader needs to verify the support result (Line 45–60 in Algorithm 4), but also all followers need to verify the support result. The follower support result verification behavior is essentially the same as the verification of the support result of the leader’s decision behavior (Line 45–60 in Algorithm 4). Each follower receives the support rate verification result (Line 3 or 5 in Algorithm 6) from other satellite nodes, and then make statistics. Specifically, if more than half of the support rate verification results received are recorded as 1, it will reply 1 to the leader, which will add 1 to the value of the FinalConfirmNumber of the leader. If not, it will reply 0 to the leader, it will decrement the value of the FinalRefuseNumber of the leader by 1.

### 3.3. The Modified RIPPLE Algorithm

When there are no malicious nodes in the cluster for a long time, the leader node consensus result without failure can be re-elected for a long time, which results in the load imbalance of the computing resource., The number of leader nodes is increased in the RIPPLE algorithm to solve the above load imbalance problem. The leader group is established, and the nodes in the leader group act as leaders in turn by using a token passing algorithm [45,46]. In some extreme cases, the scope of the leader group can be extended to the whole cluster, which means that all nodes will take turns to be leaders, according to certain rules.

The RIPPLE algorithm is a consistency algorithm suitable for a public chain. The core technology of the traditional RIPPLE algorithm, which is called the establishment of the leader group, is used for reference based on the PBFT algorithm (mentioned in Section 3.2) in this section. Compared with the RAFT and PBFT algorithms, there is one more node identity which is called the T-leader in the RIPPLE algorithm. The T-leader is the abbreviation for a temporary leader. The leader group is composed of one leader and several T-leaders. The nodes in the leader group act as leader in turn by using a token passing algorithm. When a node in the leader group is successfully selected to be the leader, other nodes in the leader group automatically translate their identities to T-leaders. It should be noted that the node in the leader group can be replaced by the node whose current identity is as a follower by using the election mechanism of the RAFT algorithm, according to the actual situation.

Main workflow of the RIPPLE algorithm can be described as the following steps (Figure 5).

Step 1: Initialization. The whole satellite cluster is composed of one leader, several T-leaders and followers. When node A, in the leader group, is selected to be the leader, the other nodes in the leader group (T-leader B, C, D) can be regarded as followers.

Step 2: Pre-release of the mission blocks. The leader releases the mission blocks to all T-leaders and followers, in the form of a broadcast. The mission allocation results for all nodes are stored in the mission blocks.

Step 3: Three-stage consensus process. Each node, except the leader, carries out a block verification, a support rate verification and a support result verification, respectively, after receiving the mission blocks released by the leader. Following this, each satellite node releases its verification result of the support result to the leader. The detailed three-stage consensus process can be seen in Section 3.2.

Step 4: Release and execution of the missions. The leader receives and analyzes the verification result of the support result released by all other followers. It is considered that the current mission allocation result released by the leader will be successfully validated if more than half of the verification results of the support results are recorded as 1. In this case, the leader will officially release the mission allocation result, and then each satellite node will execute the mission according to the mission allocation results. It is considered that the verification of the current mission allocation result released by the leader will fail if more than half of the verification results of the support results are recorded as 0. In this case, the missions allocated by the leader will not be executed and a new leader will be elected in the satellite cluster.

Step 5: Token passing process. Once the current leader completes a mission release, the next node B in the leader group will be selected as the new leader, according to the rules of the token passing algorithm, which is responsible for the release of the next mission (Step 2–5).

It should be noted that the support rate of the leader is also considered in the RIPPLE algorithm, which is very similar to the PBFT algorithm. If the support rate of the leader is low, it will be kicked out of the leader group and become a follower. At the same time, a follower will be selected to join the leader group. Hence, the security of the overall system will be effectively improved.

The improved RIPPLE algorithm makes all satellites reach a consensus through three behaviors: the mission origin behavior, the leader decision behavior and the follower behavior. The pseudo code of the above three behaviors can be shown in Algorithms 1 and 5–8, respectively. 

The mission origin behavior of the modified RIPPLE algorithm is the same as that of the DDPOS algorithm, which can be seen in Algorithm 1.
**Algorithm 8.** The leader decision behavior of the modified RIPPLE algorithm**Input:** The set of all of the satellite nodes, the total number of satellite nodes N_S_
**Output:** Consensus result **1: Switch** ucStep **do**
**2: Case** 0 **3:** The Case 0 of the leader decision behavior of the modified RIPPLE algorithm is the same as that of the PBFT algorithm **4: Case** 1 **5:** The Case 1 of the leader decision behavior of the modified RIPPLE algorithm is the same as that of the PBFT algorithm **6: Case** 2 **7:** The Case 2 of the leader decision behavior of the modified RIPPLE algorithm is the same as that of the PBFT algorithm **8: Case** 3 **9:** The Case 3 of the leader decision behavior of the modified RIPPLE algorithm is the same as that of the PBFT algorithm **10: Case** 4 **11: If** FinalConfirmNumber * 2 > N_S_
**then**
**12:    If** FinalConfirmNumber > FinalRefuseNumber **then**
**13:      ** The leader releases the mission allocation results officially to all satellite nodes **14:      ** The next node in the leader group will be selected to be the new leader according to the rules of token passing algorithm, and the current leader is changed to be the T-leader **15:      ** Restart the receive timer **16:    Else If**
**17:** The mission allocation fails, the next node in the leader group will be selected to be the new leader according to the rules of token passing algorithm, and the current leader will be replaced by a follower through the leader election mechanism of the RAFT algorithm **18:    End If**
**19:      ** FinalConfirmNumber = 0 **20:      ** FinalRefuseNumber = 0 **21:      ** Restart the receive timer **22:      ** ucStep = 0 **23: End If**
**24:    If** The receive timer times out **than**
**25:      ** The mission allocation fails, the next node in the leader group will be selected to be the new leader according to the rules of token passing algorithm, and the current leader will be replaced by a follower through the leader election mechanism of the RAFT algorithm **26:      ** FinalConfirmNumber = 0 **27:      ** FinalRefuseNumber = 0 **28:      ** Restart the receive timer **29:      ** ucStep = 0 **30:    Break**
**31: End If**
**32: Break**
**33: End Switch**

The leader decision behavior of the modified RIPPLE algorithm can be seen in Algorithm 7. It should be noted that the leader decision behavior is also a periodic behavior, which needs to be performed in each simulation step. The initial value of the ucStep is 0. There are only two differences between the leader’s decision behavior of the modified RIPPLE algorithm and the leader’s decision behavior of the modified PBFT algorithm. First, is the establishment of the leader group and second, is the token passing algorithm, which can be seen in Lines 14, 17 and 25, in Algorithm 8.

The follower behavior of the RIPPLE algorithm is essentially the same as that of the PBFT algorithm, which can be seen in Algorithms 5–7.

### 3.4. The Modified DPOS Algorithm

It is necessary to reduce the information interaction between nodes in the cluster under the situation that the repeated negotiation between nodes is required due to the frequent negotiation errors that occur in the cluster, and this is the advantage of the DPOS algorithm. The leader group is established in the DPOS algorithm. Prior to the release of the mission block, the leader group needs to reach an agreement, and all leader nodes need to sign and verify the pre-released mission block. Only the verified mission blocks can be released in the cluster.

The DPOS algorithm is a consistency algorithm suitable for a public chain. The core technology of the traditional RIPPLE algorithm, which is called the signature verification mechanism, is used for the reference based on the PBFT algorithm (mentioned in Section 3.2) in this section. The leader group is established, which is composed of one leader and several T-leaders. The mission block needs to be verified by each T-leader in the leader group by using the method of signature verification before it can be officially released to all followers.

The main workflow of the DPOS algorithm can be described in the following steps (Figure 6).

Step 1: Initialization. The whole satellite cluster is composed of one leader, several T-leaders and followers, which is similar to the initialization of the RIPPLE algorithm. In the leader group, the leader is responsible for releasing the missions, and the T-leaders are responsible for signing the verification of the mission block.

Step 2: Signature verification in the leader group. The leader releases the mission block to all T-leaders. The mission allocation results for all satellites are stored in the mission block. If the T-leaders pass the verification of the mission block, it will return its signature to the leader.

Step 3: Release of the mission. The current mission allocation results will be considered feasible if the leader receives more than half of the T-leader’s signatures. In this case, The leader releases the mission blocks to all followers in the form of a broadcast. 

Step 4: Three-stage consensus process. Each node, except the leader, carries out a block verification, a support rate verification and a support result verification, respectively, after receiving the mission blocks released by the leader. Following this, each satellite node releases its verification results of the support results to the leader. The detailed three-stage consensus process can be seen in Section 3.2.

Step 5: Execution of the missions. The leader receives and analyzes the verification results after the three-stage consensus process (See details in Section 3.2). Upon a successful verification, the leader will officially release the mission allocation results, and then each satellite node will execute the mission according to the mission allocation results.

It should be noted that the support rate of the leader is also considered in the DPOS algorithm, which is very similar to the PBFT algorithm. If the support rate of the leader is low, it will be kicked out of the leader group and become a follower. At the same time, a follower will be selected to join the leader group.

The modified DPOS algorithm makes all satellites reach a consensus through three behaviors: the mission’s origin behavior, the leader’s decision behavior, the T-leader’s signature behavior and the follower’s behavior. The pseudo code of the above three behaviors can be shown in Algorithms 1, 5–7, 9 and 10, respectively.

The mission’s origin behavior of the modified DPOS algorithm is the same as that of the DDPOS algorithm, which can be seen in Algorithm 1.
**Algorithm 9.** The leader decision behavior of the modified DPOS algorithm**Input:** The set of all satellite nodes, the total number of satellite nodes N_S_
**Output:** Consensus result **1: Switch** ucStep **do**
**2: Case** 0 **3: If** the decision timer does not time out **then**
**4:   Break**
**5: End If**
**6:  ** Restart the decision timer **7: If** the leader does not receive any desired missions **then**
**8:   Break**
**9: End If**
**10:   ** The leader and all T-leaders allocate missions according to the desired mission set **11:   ** The leader writes the mission allocation results into the mission block // Signature verification in the leader group **12:** The leader broadcasts the mission block to all other T-leaders through inter-satellite communication **13:** The leader assesses whether the missions assigned to itself can be executed **14: If** the leader can complete the missions assigned to itself **then**
**15:   ** SignatureConfirmNumber = 1 **16:   ** SignatureRefuseNumber = 0 **17: Else If**
**18:   ** SignatureConfirmNumber = 0 **19:   ** SignatureRefuseNumber = 1 **20: End If**
**21:   ** ucStep = 1 **22: Break**
**23: Case** 1 **24: If** SignatureConfirmNumber * 2 > N_S_ || The receive timer times out **then**
**25:    If** SignatureConfirmNumber > SignatureRefuseNumber **then**
**26:       Continue**
**27:    Else If**
**28:      ** The mission allocation fails, the next node in the leader group will be selected to be the new leader **29:** according to the rules of the token passing algorithm, and the current leader **30:** will be replaced by a follower through the leader election mechanism of the **31:** RAFT algorithm **32:      ** ucStep = 0 **33:      ** SignatureConfirmNumber = 0 **33:      ** SignatureRefuseNumber = 0 **34:    Break**
**35: End If**
**36:** // Verification of the block **37:** The leader broadcasts the mission block to other followers through inter-satellite communication **38:** Restart the receive timer **39:** The leader assesses whether the missions assigned to itself can be executed **40:**
**If** the leader can complete the missions assigned to itself **then**
**41:   ** PrepareConfirmNumber = 1 **42:   ** PrepareRefuseNumber = 0 **43: Else If**
**44:   ** PrepareConfirmNumber = 0 **45:   ** PrepareRefuseNumber = 1 **46: End If**
**47:   ** ucStep = 2 **48:   ** Restart the receive timer **49:   ** SignatureConfirmNumber = 0 **50:   ** SignatureRefuseNumber = 0 **51: End If**
**52: Break**
**53: Case** 2 **54: //** Verification of the support rate **55: If** PrepareConfirmNumber * 2 > N_S_ || The receive timer times out **then**
**56:    If** PrepareConfirmNumber > PrepareRefuseNumber **then**
**57:      ** The support rate verification result of the leader is recorded as 1 **58:      ** CommitConfirmNumber = 1 **59:      ** CommitRefuseNumber = 0 **60:    Else If**
**61:      ** The support rate verification result of the leader is recorded as 0 **62:      ** CommitConfirmNumber = 0 **63:      ** CommitRefuseNumber = 1 **64:    End If**
**65:      ** The leader broadcasts its support rate verification result to other satellite nodes through inter-satellite communication **66:      ** PrepareConfirmNumber = 0 **67:      ** PrepareRefuseNumber = 0 **68:      ** Restart the receive timer **69:      ** ucStep = 3 **70: End If**
**71: Break**
**72: Case 3**
**73:** // Verification of the support result **74: If** CommitConfirmNumber * 2 > N_S_ || The receive timer times out **then**
**75:    If** CommitConfirmNumber > CommitRefuseNumber **then**
**76:      ** The verification result of the support result of the leader is recorded as 1 **77:      ** FinalConfirmNumber = 1 **78:      ** FinalRefuseNumber = 0 **79:    Else If**
**80:      ** The verification result of the support result of the leader is recorded as 0 **81:      ** FinalConfirmNumber = 0 **82:      ** FinalRefuseNumber = 1 **83:    End If**
**84:      ** CommiteComfirmNumber = 0 **85:      ** CommitRefuseNumber = 0 **86:      ** Restart the receive timer **87:      ** ucStep = 4 **88: End If**
**89: Break**
**90: Case** 4 **91: If** FinalConfirmNumber * 2 > N_S_
**then**
**92:    If** FinalConfirmNumber > FinalRefuseNumber **then**
**93:      ** The leader releases the mission allocation results officially to all satellite nodes **94:      ** The next node in the leader group will be selected to be the new leader according to the rules of the token passing algorithm, and the current leader is changed to be the T-leader **95:      ** Restart the receive timer **96:    Else If**
**97:      ** The mission allocation fails, the next node in the leader group will be selected to be the new leader according to the rules of the token passing algorithm, and the current leader will be replaced by a follower through the leader election mechanism of the RAFT algorithm **98:    End If**
**99:      ** FinalConfirmNumber = 0 **100:    ** FinalRefuseNumber = 0 **101:    ** Restart the receive timer **102:    ** ucStep = 0 **103: End If**
**104: If** The receive timer times out **than**
**105: ** The mission allocation fails, the next node in the leader group will be selected to be the new leader, according to the rules of the token passing algorithm, and the current leader will be replaced by a follower through the leader election mechanism of the RAFT algorithm **106: ** FinalConfirmNumber = 0 **107: ** FinalRefuseNumber = 0 **108: ** Restart the receive timer **109: ** ucStep = 0 **Break**
**End If**
**Break**
**End Switch**

The leader decision behavior of the modified DPOS algorithm can be seen in Algorithm 9. It should be noted that the leader’s decision behavior is also a periodic behavior, which needs to be performed in each simulation step. The initial value of the ucStep is 0.

The T-leader signature behavior of the modified DPOS algorithm can be seen in Algorithm 10. Once the leader broadcasts the mission block to all other T-leaders through inter-satellite communication (Line 12 in Algorithm 9), all T-leaders need to perform the T-leader’s signature behavior. Specifically, if the T-leader returns the mission block with its signature (Line 4 in Algorithm 10), it will add 1 to the value of the SignatureConfirmNumber. If not, the value of the SignatureConfirmNumber will be decremented by 1.
**Algorithm 10.** The T-leader signature behavior**Input:** T-leader i **Output:** The mission block with signature **1:** T-leader i receives the mission block broadcasted by the leader **2:** T-leader i compares its mission allocation results with that of the leader **3: If** There is no difference between the two mission allocation results **than**
**4:  ** T-leader i returns the mission block with its signature **5: Else If**
**6:   Return**
**7: End If**

The follower’s behavior of the DPOS algorithm is essentially the same as that of the PBFT algorithm, which can be seen in Algorithms 5–7.

## 4. Results

In this paper, some satellites in partial orbits of a Walker constellation are selected for the simulation verification. The satellite cluster of 20 satellites is selected to compare the effects of the different consistency algorithms. The orbit parameters of all satellites are shown in Table 1.

### 4.1. Resource Occupancy Comparison of the Different Algorithms

Firstly, the planner component simulation program, including the algorithms which we discussed in this paper, is compiled and tested on the embedded platform. The results are shown in Figure 7.

The code size of the five algorithms, that we discussed in this paper, can be compared from the compilation results. It can be seen from Table 2 that the code size of the RAFT algorithm is 91.616 Kb. The code size comparison of these five algorithms is shown in Table 2. The remaining four algorithms are developed based on the core technology of the RAFT algorithm. Hence, there is little difference in the code quantity between the five algorithms, which can be seen in Table 2.

These five different algorithms are used for a mission allocation test. Upon completing one mission allocation, the computing and communication resources consumption of all satellites through each algorithm are calculated. The computing resources are described by a computer crystal oscillator number which is collected by the QueryPerformanceCounter function of the Windows Platform before and after the mission negotiation. In this paper, the computing resources are expressed in the total computation. The communication resources are archived each time data is sent, which are expressed in the total bandwidth occupation. The computing and communication resources occupation of these five different algorithms can be seen in Table 2.

It can be seen from Table 2 that there is little difference in the computing resources occupation between RAFT and PBFT. Compared with RAFT, the computing resources occupation of both RIPPLE and DPOS are significantly larger, which increase by 6–8 times. However, compared with RIPPLE and DPOS, the computing resources occupation of DDPOS is much lower, which is of the same order of magnitude as the computing resources occupation of RAFT. Moreover, the communication resources occupation of these five algorithms are of the same order of magnitude. Surprisingly, it can be seen from Table 2 that compared with PBFT, RIPPLE and DPOS, the communication resources occupation of DDPOS is significantly lower, which is similar to the communication resources occupation of RAFT. It can be seen from Table 2 that the overall resource occupation of the DDPOS algorithm is lower than that of the other four algorithms in the case of malicious nodes. Especially compared with the RAFT algorithm, the total computation and total bandwidth occupation of the DDPOS algorithm have decreased by 67% and 75%, respectively. It can be also seen from Table 2, that all algorithms except the RAFT algorithm can effectively resist the attack of malicious nodes. Compared with the other four algorithms, the advantage of the DDPOS algorithm is the high freedom of selection. Satellites in the cluster can enter and leave freely, according to its actual situation, and each cluster can freely choose the appropriate algorithm among the four algorithms of RAFT, PBFT, RIPPLE and DPOS. Twenty satellites are divided into two satellite clusters through the DDPOS algorithm. The number of satellites contained in a single satellite cluster is greatly reduced, which will effectively reduce the difficulty of the decision-making in the satellite cluster. Hence, the computing and communication resources occupation of the whole satellite cluster can be effectively reduced.

### 4.2. Performance Comparison of the Different Algorithms

The resources occupation of the algorithm is important, but so is the security performance of the algorithm. For the decentralized consistency algorithm, the ability to resist malicious nodes is extremely important. In this section, six malicious nodes are set, which are satellite ID 4, 6, 8, 14, 16 and 18, respectively. The effects of setting malicious nodes or not on computing and communication resources occupation through each algorithm are compared. It should be noted that malicious nodes are mainly responsible for transmitting error information to other nodes in this paper, and the leader is responsible for making all nodes in the satellite cluster reach a consensus.

#### 4.2.1. RAFT

In the RAFT algorithm, Satellite 1 is selected as the leader and other satellites are followers. The results of the total computation and total bandwidth occupation of the satellite cluster with or without malicious nodes are shown in Table 3. The distribution of the total computation and total bandwidth occupation of the satellite cluster with or without malicious nodes are shown in Figure 8 and Figure 9.

For the satellite cluster without malicious nodes, the total computation is about 437.9 K. The computation used by the leader is about 401.4 K, which accounts for about 92% of the total computation. The computation used by each follower is less than 1%. At the same time, the total bandwidth occupation is about 304.7 Kb. The bandwidth occupation used by the leader is about 7.66 Kb. which accounts for about 3% of the total bandwidth occupation. The bandwidth occupation used by each follower ranges from 4% to 6%.

Compared with the satellite cluster without malicious nodes, the total computation and the total bandwidth occupation of the satellite cluster with malicious nodes are much larger, which are 2067 K and 1336.1 Kb, respectively. Moreover, the computation used by the leader is about 2038.6 K, which accounts for about 99% of the total computation. The computation used by each follower is less than 1%. At the same time, the total bandwidth occupation is about 1336.1 Kb. The bandwidth occupation used by the leader is about 1039 Kb. which accounts for about 99% of the total bandwidth occupation. The bandwidth occupation used by each follower is less than 1%. The above results show that the negotiation process is abnormal due to the existence of the malicious nodes. The ability of the RAFT algorithm against malicious nodes is insufficient.

#### 4.2.2. PBFT

In the PBFT algorithm, Satellite 1 is selected as the leader and other satellites are followers. The results of the total computation and the total bandwidth occupation of the satellite cluster with or without malicious nodes are shown in Table 4. The distribution of the total computation and the total bandwidth occupation of the satellite cluster with or without malicious nodes are shown in Figure 10 and Figure 11.

For the satellite cluster without malicious nodes, the total computation is about 382.3 K. The computation used by the leader is about 351.7 K, which accounts for about 92% of the total computation. The computation used by each follower is less than 1%. At the same time, the total bandwidth occupation is about 594.9 Kb. The bandwidth occupation used by the leader is about 15.18 Kb. which accounts for about 3% of the total bandwidth occupation. The bandwidth occupation used by each follower ranges from 4% to 6%.

Compared with the satellite cluster without malicious nodes, the total computation and the total bandwidth occupation of the satellite cluster with malicious nodes are similar, which are 407.8 K and 594.9 Kb, respectively. The negotiation can be successfully completed through the PBFT algorithm. Moreover, the computation used by the leader is about 368.1 K, which accounts for about 90% of the total computation. The computation used by each follower is less than 1%. At the same time, the total bandwidth occupation is about 594.9 Kb. The bandwidth occupation used by the leader is about 15.18 Kb. which accounts for about 3% of the total bandwidth occupation. The bandwidth occupation used by each follower ranges from 5% to 6%. The above results show that the interference of malicious nodes can be resisted effectively through the PBFT algorithm.

#### 4.2.3. RIPPLE

In the RIPPLE algorithm, satellites 1–5 are selected as the leaders and other satellites are followers. The satellites 1–5 are gathered to form the leader group. The results of the total computation and the total bandwidth occupation of the satellite cluster with or without malicious nodes are shown in Table 5. The distribution of the total computation and the total bandwidth occupation of the satellite cluster with or without malicious nodes are shown in Figure 12 and Figure 13.

For the satellite cluster without malicious nodes, the total computation is about 2920.7 K. The computation used by the leader is about 429 K, which accounts for about 15% of the total computation. The computation used by each follower ranges from 1% to 6%. At the same time, the total bandwidth occupation is about 439.2 Kb. The bandwidth occupation used by the leader is about 24.8 Kb. which accounts for about 6% of the total bandwidth occupation. The bandwidth occupation used by each follower ranges from 3% to 6%.

Compared with the satellite cluster without malicious nodes, the total computation and the total bandwidth occupation of the satellite cluster with malicious nodes are similar, which are 3249 K and 656 Kb, respectively. The negotiation can be successfully completed through the RIPPLE algorithm. Moreover, the computation used by the leader is about 497.5 K, which accounts for about 15% of the total computation. The computation used by each follower ranges from 1% to 5%. At the same time, the total bandwidth occupation is about 656 Kb. The bandwidth occupation used by the leader is about 35.6 Kb. which accounts for about 5% of the total bandwidth occupation. The bandwidth occupation used by each follower ranges from 4% to 6%. The above results show that the interference of malicious nodes can be resisted effectively through the RIPPLE algorithm.

#### 4.2.4. DPOS

In the DPOS algorithm, satellites 1–5 are selected as the leaders and other satellites are followers. The satellites 1–5 are gathered to form the leader group. The results of the total computation and the total bandwidth occupation of the satellite cluster with or without malicious nodes are shown in Table 6. The distribution of the total computation and the total bandwidth occupation of the satellite cluster with or without malicious nodes are shown in Figure 14 and Figure 15.

For the satellite cluster without malicious nodes, the total computation is about 4462.9 K. The computation used by the leader is about 141.1 K, which accounts for about 3% of the total computation. The computation used by each follower ranges from 1% to 15%. At the same time, the total bandwidth occupation is about 476.8 Kb. The bandwidth occupation used by the leader is about 7.6 Kb. which accounts for about 2% of the total bandwidth occupation. The bandwidth occupation used by each follower ranges from 2% to 8%.

Compared with the satellite cluster without malicious nodes, the total computation and the total bandwidth occupation of the satellite cluster with malicious nodes are similar, which are 4502.9 K and 476.8 Kb, respectively. The negotiation can be successfully completed through the DPOS algorithm. Moreover, the computation used by the leader is about 143.2 K, which accounts for about 3% of the total computation. The computation used by each follower ranges from 4% to 15%. At the same time, the total bandwidth occupation is about 476.8 Kb. The bandwidth occupation used by the leader is about 7.6 Kb. which accounts for about 2% of the total bandwidth occupation. The bandwidth occupation used by each follower ranges from 2% to 8%. The above results show that the interference of malicious nodes can be resisted effectively through the DPOS algorithm.

#### 4.2.5. DDPOS

In the DDPOS algorithm, satellites 1–10 are gathered to form the first leader group, and satellites 11–20 are gathered to form the second leader group. The satellite 1 and 11 are selected as the leaders and other satellites are followers. The PBFT algorithm is selected for the negotiation in the satellite cluster and the RAFT algorithm is used for the negotiation among satellite clusters. The results of the total computation and the total bandwidth occupation of the satellite cluster with or without malicious nodes are shown in Table 7. The distribution of the total computation and the total bandwidth occupation of the satellite cluster with or without malicious nodes are shown in Figure 16 and Figure 17.

For the satellite cluster without malicious nodes, the total computation is about 686.4 K. The computations used by the two leaders are about 185.3 K and 441.6 K, which account for about 27% and 64% of the total computation, respectively. The computation used by each follower is less than 1%. At the same time, the total bandwidth occupation is about 316 Kb. The bandwidth occupations used by both two leaders are about 18.5 Kb. which account for about 6% of the total bandwidth occupation. The bandwidth occupation used by each follower ranges from 3% to 6%.

Compared with the satellite cluster without malicious nodes, the total computation and the total bandwidth occupation of the satellite cluster with malicious nodes are similar, which are 680.1 K and 333.8 Kb, respectively. The negotiation can be successfully completed through the DDPOS algorithm. Moreover, the computation used by the two leaders is about 434 K, which accounts for about 64% of the total computation. The computation used by each follower is less than 1%. At the same time, the total bandwidth occupation is about 333.8 Kb. The bandwidth occupation used by the two leaders is about 18.5 Kb. which accounts for about 6% of the total bandwidth occupation. The bandwidth occupation used by each follower ranges from 3% to 6%. The above results show that the interference of malicious nodes can be resisted effectively through the DDPOS algorithm.

## 5. Discussion

In this Section, the performances of RAFT, PBFT, RIPPLE, DPOS and DDPOS are compared and analyzed. The related comparison results can be seen in Table 2. These five algorithms are analyzed from three aspects: code quantity, resource occupancy and security. 

First is the code quantity. It can be seen in Table 2 that there is little difference in code quantity between the five algorithms. Compared with the RAFT algorithm, the code quantity of the DDPOS algorithm is only increased by around 7%. This is due to the fact that the code quantity of the algorithm calculated in this paper mainly refers to the code quantity of the complete program that can be run on-board. The complete program is mainly composed of the consistency algorithm code, the mission planning algorithm code and some other basic code of mechanics and mathematics. In fact, the code quantity of the consistency algorithm only accounts for a very small part of the code quantity of the complete program. For example, the code quantity of the RAFT algorithm only accounts for around 6% of the code quantity of the complete RAFT program. 

Second is the resource occupancy. The total computation and the total bandwidth occupation of these five algorithms are compared. It can be seen in Table 2 that compared with the other four algorithms, the resource occupancy of the RAFT algorithm is minimal due to its simple algorithm structure. The resource occupation of the DDPOS and PBFT algorithms are the same order of magnitude as that of the RAFT algorithm. These results may be due to the fact that, in this experiment, DDPOS is composed of the RAFT and PBFT algorithms. In addition, compared with the RAFT algorithm, the total computation of the RIPPLE and DPOS algorithms are significantly higher, which are increased by around 517% and 919%, respectively. This is due to the fact that the establishment of the leader group and the design of the signature verification mechanism significantly increase the total computation of the whole satellite cluster. Moreover, it should be noted that although the total computation and total bandwidth occupation of the PBFT algorithm are similar to that of the DDPOS algorithm, the computation of a single satellite is greatly different between the PBFT and DDPOS algorithms, which can be clearly seen in 15, 16, 21 and 22. Compared with the PBFT algorithm, the computation of each satellite is more balanced in the DDPOS algorithm, especially for the leaders. For the satellite cluster with malicious nodes, the maximum computation of a single satellite accounts for about 90% of the total computation in the PBFT algorithm, which is an extremely high proportion. However, the maximum computation of a single satellite accounts for about 64% of the total computation in the DDPOS algorithm, which decreases significantly compared with the PBFT algorithm. This is due to the fact that the number of leaders of the DDPOS algorithm is more than that of the PBFT algorithm and multiple leaders can share the computing pressure together. It should be noted that compared with the DDPOS algorithm, the computation of each satellite is more balanced in the RIPPLE and DPOS algorithms. However, compared with the RIPPLE and DPOS algorithms, the total computation of the DDPOS is much less, which is decreased by around 79% and 85% less than that of the RIPPLE and DPOS algorithms, respectively. Hence, the RIPPLE and DPOS algorithms are not suitable for the large-scale satellite clusters.

Third is the security. As we all know, high security and low resource occupation cannot be achieved at the same time. For example, the RAFT algorithm has the lowest resource occupancy, but it cannot resist malicious nodes. The appropriate consistency algorithms can be freely chosen according to the current actual situation in the DDPOS algorithm. In particular, we can select the most suitable consistency algorithm according to the actual situation of each satellite cluster. For example, if there are no malicious nodes in the current satellite cluster, the consistency algorithm with a low security and a low resource occupation will be chosen for the satellite cluster. By contrast, if there are some malicious nodes in the current satellite cluster, the consistency algorithm with a high security and a high resource occupation will be chosen for the satellite cluster. In this experiment, The leaders are not malicious nodes in the DDPOS algorithm. Hence, the PBFT algorithm is chosen in the first layer of the DDPOS algorithm, and the RAFT algorithm is chosen in the second layer of the DDPOS algorithm. In the first layer of the DDPOS algorithm, the satellites in each satellite cluster reach a consensus through the PBFT algorithm, which can effectively resist malicious nodes. At the same time, malicious nodes can be effectively prevented from entering the second layer of the DDPOS algorithm through the PBFT algorithm, which ensures the security of the second layer of the DDPOS algorithm. Moreover, the leaders of all satellite clusters reach a consensus through the RAFT algorithm, which reduces the resource occupation of the DDPOS algorithm to a certain extent.

Hence, we believe that compared with the RAFT, PBFT, RIPPLE and DPOS algorithms, DDPOS has obvious advantages in terms of performance. We are convinced that compared with the above four algorithms, the DDPOS algorithm is more suitable for the large-scale satellite clusters.

## 6. Conclusions

The common decentralized consistency algorithms suitable for the Internet are analyzed in this paper. According to the application scopes and characteristics of each algorithm, four common consistency algorithms are selected and modified for multi-satellite autonomous mission allocation, which named RAFT, PBFT, RIPPLE and DPOS, respectively. Moreover, based on the above four modified consistency algorithms, a new double-layer decentralized consistency algorithm named DDPOS is proposed. Finally, two working conditions with and without malicious nodes are set to compare the performance of the traditional four algorithms and the new algorithm proposed in this paper.

The results show that the code quantity of the DDPOS algorithm is only increased by around 7%, more than that of the RAFT algorithm with a minimum code quantity. At the same time, the total computation and the total bandwidth occupation of the DDPOS algorithm is in the same order of magnitude as the RAFT and the PBFT algorithm. However, the RAFT and PBFT algorithms have their own shortcomings. The RAFT algorithm cannot resist malicious nodes. Compared with the DDPOS algorithm, the computation of each satellite is less balanced in the PBFT algorithm, especially for the leader. The computation of the leader accounts for about 90% of the total computation in the PBFT algorithm, which is an extremely high proportion. Moreover, compared with the DDPOS algorithm, the computation of each satellite is more balanced in the RIPPLE and DPOS algorithms. However, compared with the RIPPLE and DPOS algorithms, the total computation of DDPOS is much less, which is decreased by around 79% and 85% less than that of the RIPPLE and DPOS algorithms, respectively. Hence, compared with the traditional four algorithms, the DDPOS algorithm not only has better security and reliability, but also occupies less computing and bandwidth resources, which can integrate the advantages of the traditional four algorithms. The results show that the overall performance of the DDPOS algorithm is very high compared to the other traditional four algorithms.

## Figures and Tables

**Figure 1 sensors-22-07387-f001:**
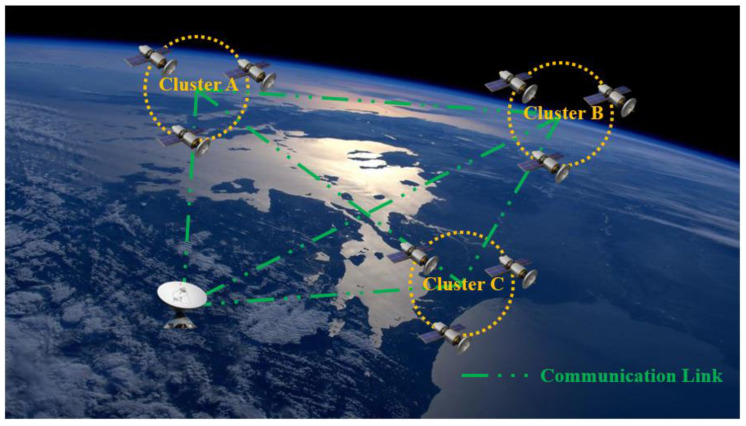
The basic decentralized communication architecture of the satellite cluster.

**Figure 2 sensors-22-07387-f002:**
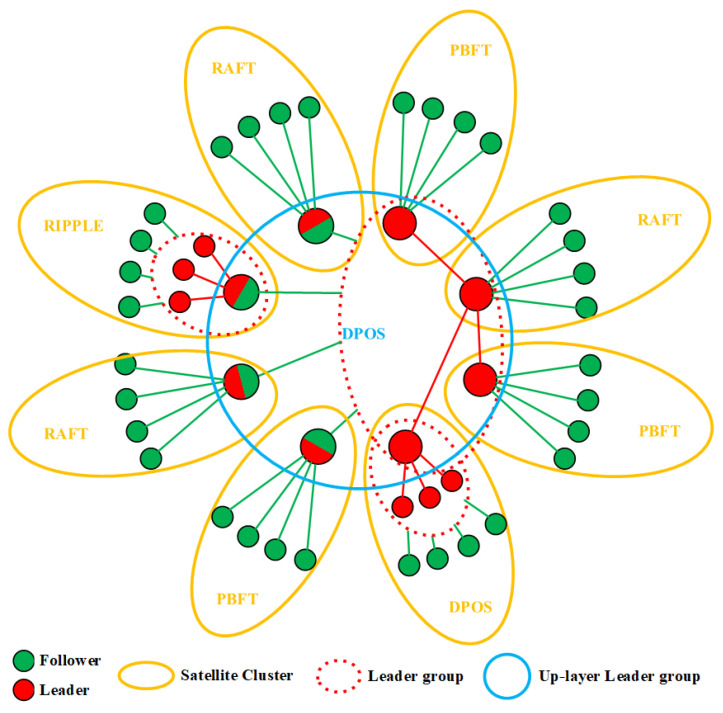
The structure diagram of the DDPOS algorithm.

**Figure 3 sensors-22-07387-f003:**
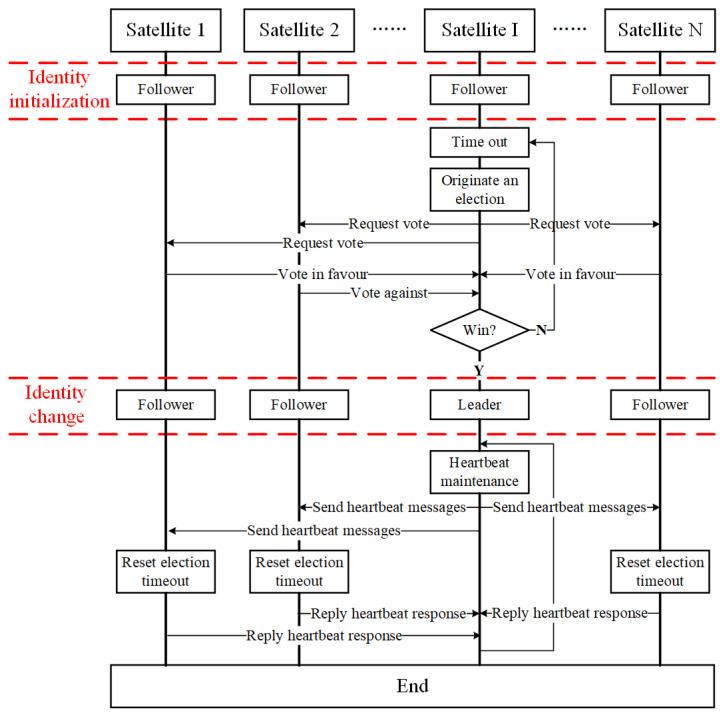
The leader election process of RAFT.

**Figure 4 sensors-22-07387-f004:**
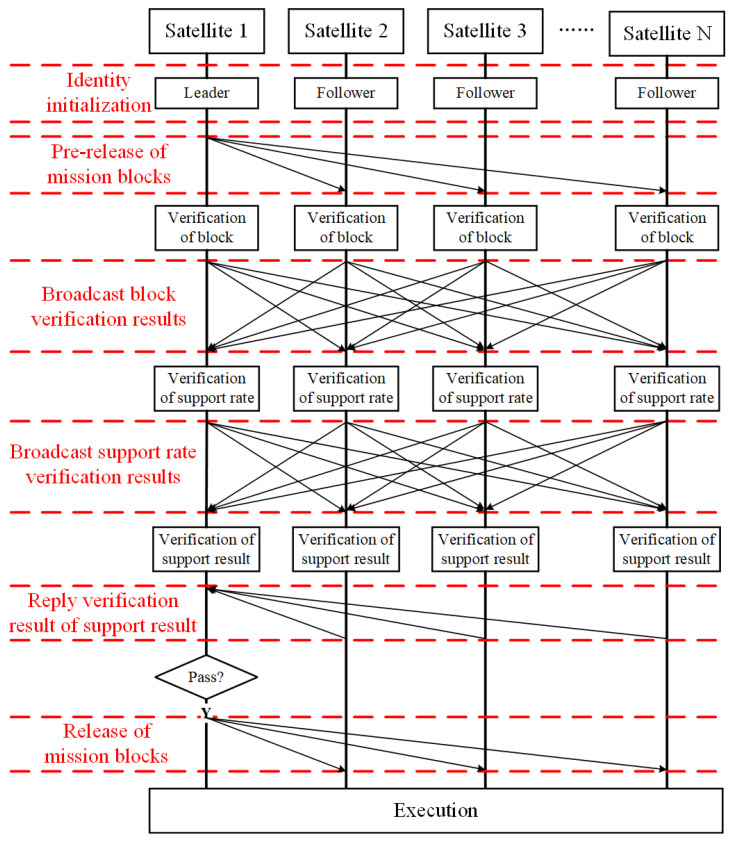
Algorithm flowchart of PBFT.

**Figure 5 sensors-22-07387-f005:**
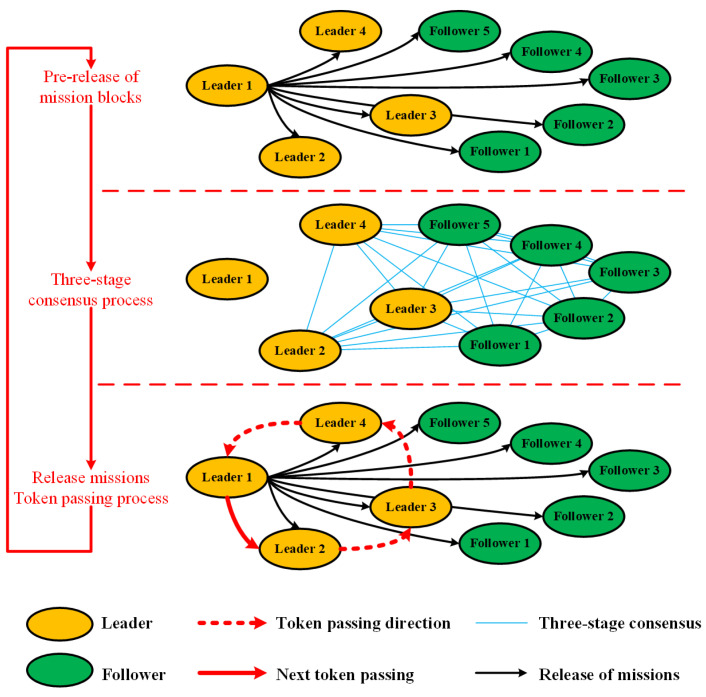
Algorithm flowchart of RIPPLE.

**Figure 6 sensors-22-07387-f006:**
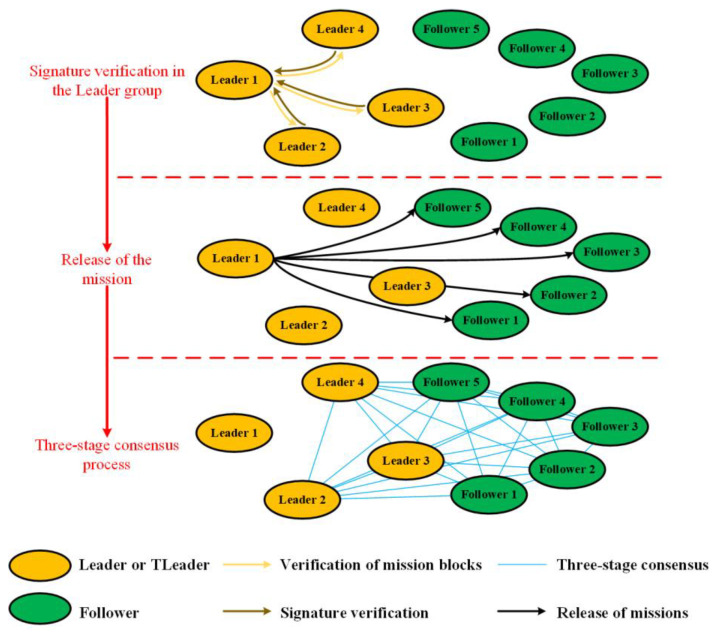
Algorithm flowchart of DPOS.

**Figure 7 sensors-22-07387-f007:**
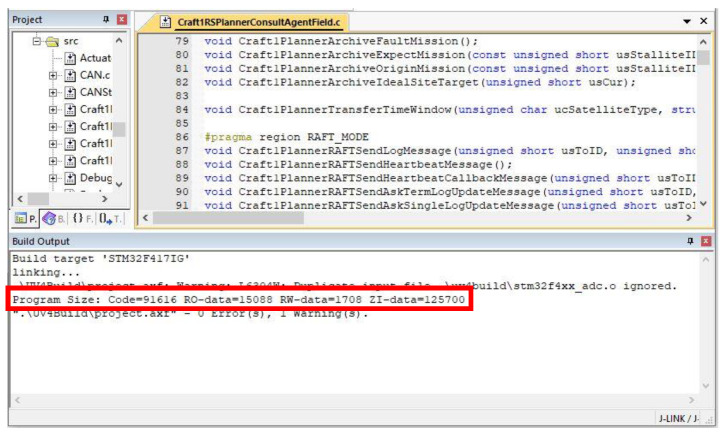
Compilation results of the planner component simulation program including the RAFT algorithm.

**Figure 8 sensors-22-07387-f008:**
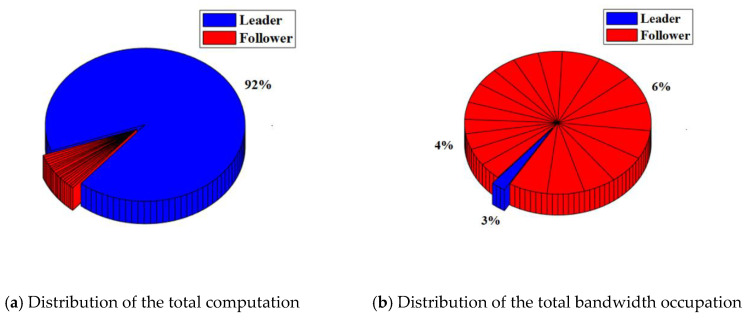
Resource occupation of the RAFT algorithm with no malicious nodes.

**Figure 9 sensors-22-07387-f009:**
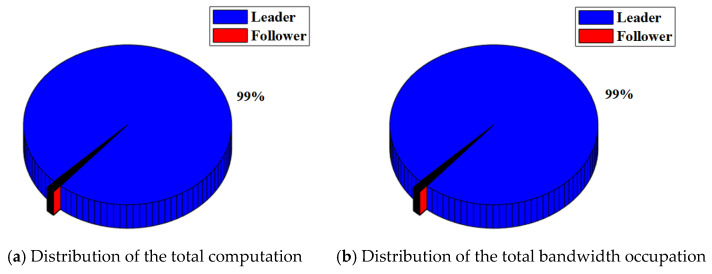
Resource occupation of the RAFT algorithm with malicious nodes.

**Figure 10 sensors-22-07387-f010:**
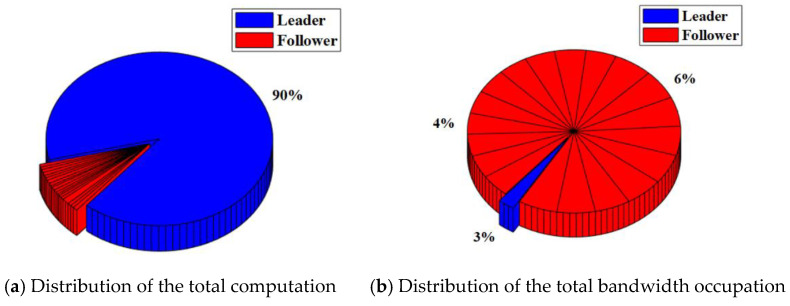
Resource occupation of the algorithm with no malicious nodes.

**Figure 11 sensors-22-07387-f011:**
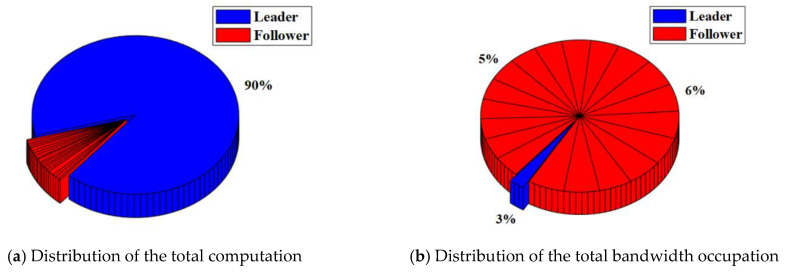
Resource occupation of the PBFT algorithm with malicious nodes.

**Figure 12 sensors-22-07387-f012:**
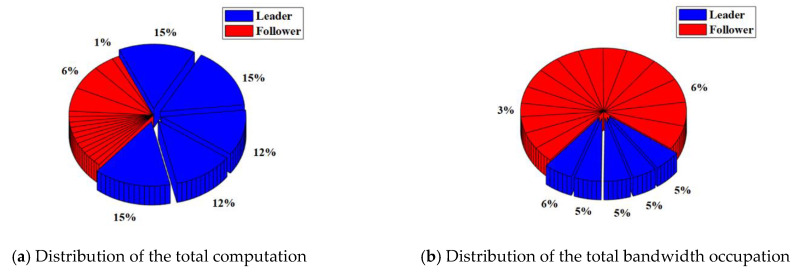
Resource occupation of the RIPPLE algorithm with no malicious nodes.

**Figure 13 sensors-22-07387-f013:**
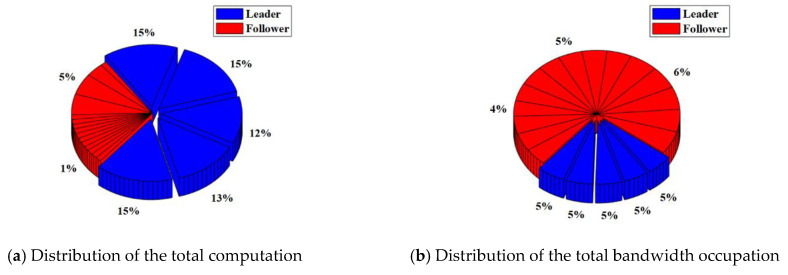
Resource occupation of the RIPPLE algorithm with malicious nodes.

**Figure 14 sensors-22-07387-f014:**
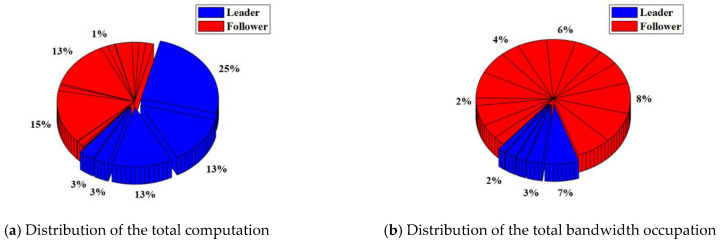
Resource occupation of the DPOS algorithm with no malicious nodes.

**Figure 15 sensors-22-07387-f015:**
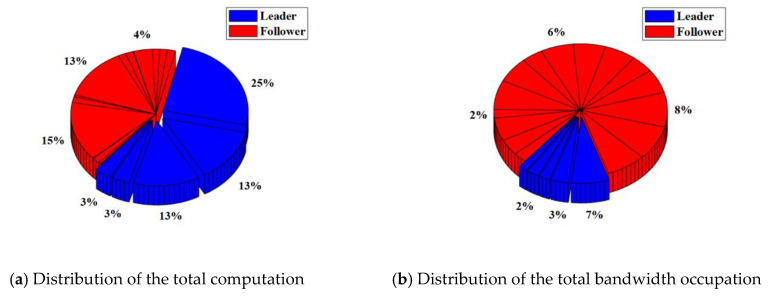
Resource occupation of the DPOS algorithm with malicious nodes.

**Figure 16 sensors-22-07387-f016:**
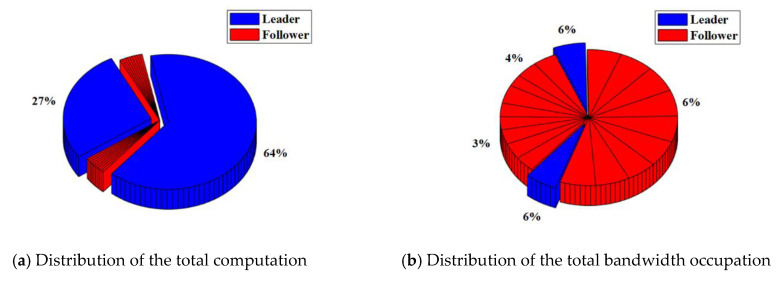
Resource occupation of the DDPOS algorithm with no malicious nodes.

**Figure 17 sensors-22-07387-f017:**
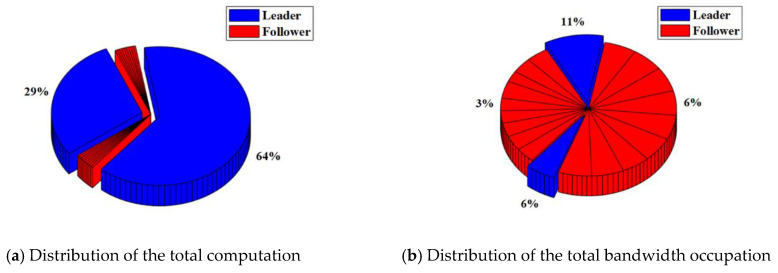
Resource occupation of the DDPOS algorithm with malicious nodes.

**Table 1 sensors-22-07387-t001:** Parameters for the satellites.

Satellite ID	Semimajor Axis	Eccentricity	Inclination	Ascending Node	Perigee Argument	True Anomaly
1	6,878,000	0	40	80	0	130
2	6,878,000	0	40	80	0	125
3	6,878,000	0	40	80	0	120
4	6,878,000	0	40	80	0	115
5	6,878,000	0	40	80	0	110
6	6,878,000	0	40	80	0	105
7	6,878,000	0	40	80	0	100
8	6,878,000	0	40	80	0	95
9	6,878,000	0	40	80	0	90
10	6,878,000	0	40	80	0	85
11	6,878,000	0	40	170	0	145
12	6,878,000	0	40	170	0	140
13	6,878,000	0	40	170	0	135
14	6,878,000	0	40	170	0	130
15	6,878,000	0	40	170	0	125
16	6,878,000	0	40	170	0	120
17	6,878,000	0	40	170	0	115
18	6,878,000	0	40	170	0	110
19	6,878,000	0	40	170	0	105
20	6,878,000	0	40	170	0	100

**Table 2 sensors-22-07387-t002:** The performance comparison of the five algorithms.

Algorithm Name	Code Quantity (Kb)	With Malicious Nodes		Without Malicious Nodes		Whether It Can Resist Malicious Nodes
		Total Computation (freq)	Total Bandwidth Occupation (Kb)	Total Computation (freq)	Total Bandwidth Occupation (Kb)	
RAFT	91.616	2067 K	1336.1	437.9 K	304.7	No
PBFT	92.353	407.8 K	594.9	382.3 K	594.9	Yes
RIPPLE	95.287	3249 K	656	2920.7 K	439.2	Yes
DPOS	95.915	4502.9 K	476.8	4462.9 K	476.8	Yes
DDPOS	98.375	680.1 K	333.8	686.4 K	316	Yes

**Table 3 sensors-22-07387-t003:** The performance of the RAFT algorithm.

	Total Computation (freq)	Total Bandwidth Occupation (Kb)
No malicious nodes	437.9 K	304.7
Malicious nodes contained	2067 K	1336.1

**Table 4 sensors-22-07387-t004:** The performance of the PBFT algorithm.

	Total Computation (freq)	Total Bandwidth Occupation (Kb)
No malicious nodes	382.3 K	594.9
Malicious nodes contained	407.8 K	594.9

**Table 5 sensors-22-07387-t005:** The performance of the RIPPLE algorithm.

	Total Computation (freq)	Total Bandwidth Occupation (Kb)
No malicious nodes	2920.7 K	439.2
Malicious nodes contained	3249 K	656.0

**Table 6 sensors-22-07387-t006:** The performance of the DPOS algorithm.

	Total Computation (freq)	Total Bandwidth Occupation (Kb)
No malicious nodes	4462.9 K	476.8
Malicious nodes contained	4502.9 K	476.8

**Table 7 sensors-22-07387-t007:** The performance of the DDPOS algorithm.

	Total Computation (freq)	Total Bandwidth Occupation (Kb)
No malicious nodes	686.4 K	316.0
Malicious nodes contained	680.1 K	333.8

## Data Availability

The data presented in this study are available on request from the corresponding author. The data are not publicly available due to trade secrets.

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
