# Peer review of "A New Double-Layer Decentralized Consistency Algorithm for the Multi-Satellite Autonomous Mission Allocation Based on a Block-Chain"

_sensors, 2022, doi:10.3390/s22197387_

Round 1
Reviewer 1 Report
I understand that the four selected typical consistency algorithms suitable for the Internet, named RAFT, PBFT, RIPPLE and DPOS, are used and modified for the negotiation of autonomous multi-satellite missions.
1. However, the modifications are explained informally, there is no formal explanation (which must be based on mathematical or computer concepts) that allows others to reproduce these modifications. If no explanations are given for patent purposes, then it is better to apply for a patent than to publish an article. If this is not the case, then the modifications must be given explicitly and using a mathematical basis.
2. Suppose a modification of any of the algorithms has a bug, how can you find it?
3. On the other hand, how can a reader put his proposal into practice or implemented it?
Note that, there is no certainty that the modifications and implementations are correct
Author Response
Reply to reviewer 1
Dear reviewer:
I am very grateful to your comments for the manuscript. According with your advice, we amended the relevant part in manuscript. All your questions were answered below.
Question 1:
The reviewer’s comment: However, the modifications are explained informally, there is no formal explanation (which must be based on mathematical or computer concepts) that allows others to reproduce these modifications. If no explanations are given for patent purposes, then it is better to apply for a patent than to publish an article. If this is not the case, then the modifications must be given explicitly and using a mathematical basis.
The authors’ Answer: Thanks for your kind suggestion. We really don't explain the modifications formally in the manuscript before. Now, we have supplemented relevant descriptions and the pseudo codes of these four algorithms in Section 2 and 3 of the revised manuscript to let readers understand more clearly how these four algorithms are modified.
Question 2:
The reviewer’s comment: Suppose a modification of any of the algorithms has a bug, how can you find it?
The authors’ Answer: Thank you for your question. First, the four original consistency algorithms have been successfully applied to the Internet, and their algorithmic logic all have been tested successfully. In this manuscript, the basic frameworks of four modified consistency algorithms have not been changed, which can be seen in section 3 for details. Second, the perfect code test specifications are established in our laboratory. Our laboratory has built perfect code test specifications for digital satellite software and codes. For example, testing for code is divided into static testing and dynamic testing. We generally use breakpoint testing method to find and solve bugs. If you are interested, we can send you the code test specifications of our laboratory. Third, we have carried out a lot of tests for the same working condition, and the test results show that the data very stable.
Question 3:
The reviewer’s comment: On the other hand, how can a reader put his proposal into practice or implemented it?
The authors’ Answer: Thank you for your question. In this manuscript, the DDPOS algorithm and four modified consistency algorithms have been successfully applied to on-board mission planning. Besides, the above five algorithms can be also applied to the mission planning in any field such as UAVs, robots and so on. We have supplemented the pseudo codes and relevant descriptions of all consistency algorithms in section 2 and 3. It is now easier for readers to understand how the algorithms are modified. If readers have other proposals, they can easily modify the existing pseudo codes.
I'm sorry that the modifications of all the algorithms involved in this manuscript have not been described clearly before. Now, we have supplemented relevant descriptions and the pseudo codes of all algorithms, which can be seen in section 2 and 3. Especially for the modifications of the algorithms, we have described them in detail at the beginning of section 3.
In addition to the above modifications to the manuscript, in order to let readers better understand our manuscript, we have also made three modifications to the manuscript. First is that we optimized the framework of the article and deleted some repetitive descriptions. There are 6 sections in the revised manuscript. Section 1 is “Introduction”. Section 2 is “The new double-layer decentralized consistency algorithm (DDPOS)”. Section 3 is “Four modified common consistency algorithms selected for the DDPOS”. Section 4 is “Results”. Section 5 is “Discussion”. Section 6 is “Conclusion”. Part content of the section “Background and related work” was deleted and another part was put into the section “Introduction”. We rename the names of some sections to avoid ambiguity. Second is that in order to make it easier for readers to compare the performance of each algorithm, figure 13, 14 and 15 were removed and their information were summarized in Table 2. Third is that we supplemented the section “Discussion” and modified relevant descriptions of section “Conclusion”.
Thank you for your suggestions. All your suggestions are very important, which have important guiding significance for my thesis writing and scientific research!
Reviewer 2 Report
The manuscript “A new double-layer decentralized consistency algorithm for multi-satellite autonomous mission allocation based on block chain” complies the journal scope since it presents a novel and robust approach to guarantee secure telecommunications between satellites by avoiding collapse when the master node is attacked by malicious nodes. The manuscript is clear, describes very well the problem and presents relevant and actualized bibliography. The methodology is robust and well described but it is excessive long and it is presented mixed with background and results sections. Following recommendations should be taken into account in order to improve the manuscript:
The section “Background and related work” should be removed and that information should be put in “Introduction”. Figures from “Background and related work” should be removed since the reader can find this information in the reference of each algorithm. The section ‘Modification of four common consistency algorithms” should be renamed as “Methodology” since the authors explain all the steps they have followed to perform algorithms modification. Some information from
from sections 2.1 to 2.4 could be introduced if necessary in this section. The section “A new double-layer decentralized consistency algorithms based on the above four traditional consistency algorithm” should renamed as “Results” section, since the author show there their science product. Inside Results section there can be two subsections: “New double-layer decentralized consistency algorithms” and “Simulation for different experimental configurations”. Figure 13, 14 and 15 could be removed and their information could be summarized in a Table. Finally, a separate “Discussion” section is recommended. Conclusion section is consistent with the results obtained.
The manuscript should be accepted for publication with these minor corrections stated above.
Author Response
Reply to reviewer 2
Dear reviewer:
I am very grateful to your comments for the manuscript. According with your advice, we amended the relevant part in manuscript. All your questions were answered below.
Question 1:
The reviewer’s comment: The section “Background and related work” should be removed and that information should be put in “Introduction”. Figures from “Background and related work” should be removed since the reader can find this information in the reference of each algorithm.
The authors’ Answer: Thanks for your kind suggestion. Now we have put the contents of the section “Background and related work” in the section “Introduction”. At the same time, all figures from “Background and related work” are removed. Besides, the order of relevant references was also changed.
Question 2:
The reviewer’s comment: The section ‘Modification of four common consistency algorithms” should be renamed as “Methodology” since the authors explain all the steps they have followed to perform algorithms modification.
The authors’ Answer: Thanks for your kind suggestion. Now the section ‘Modification of four common consistency algorithms” is renamed as “Four modified common consistency algorithms selected for the DDPOS” in order to in order to better understand the contents of this section. Besides, we have supplemented relevant descriptions and the pseudo codes of these four algorithms in Section 3 of the revised manuscript to let readers understand more clearly how these four algorithms are modified.
Question 3:
The reviewer’s comment: Some information from sections 2.1 to 2.4 could be introduced if necessary in this section.
The authors’ Answer: Thanks for your kind suggestion. After your reminder, we realize that the contents of sections 2.1-2.4 were duplicated with those of section 3. Hence, we deleted the contents of sections 2.1-2.4.
Question 4:
The reviewer’s comment: The section “A new double-layer decentralized consistency algorithms based on the above four traditional consistency algorithm” should renamed as “Results” section, since the author show there their science product. Inside Results section there can be two subsections: “New double-layer decentralized consistency algorithms” and “Simulation for different experimental configurations”. Figure 13, 14 and 15 could be removed and their information could be summarized in a Table.
The authors’ Answer: Thanks for your kind suggestion. According to your suggestion, we have made some corresponding modifications, which can be seen in section 2 and 4. In addition, in order to let readers better understand the new method proposed in this manuscript, we made some changes to the framework of the manuscript according to the idea of total score structure. There are 6 sections in the revised manuscript. Section 1 is “Introduction”. Section 2 is “The new double-layer decentralized consistency algorithm (DDPOS)”. Section 3 is “Four modified common consistency algorithms selected for the DDPOS”. Section 4 is “Results”. Section 5 is “Discussion”. Section 6 is “Conclusion”. The DDPOS algorithm is composed of any combination of modified RAFT, modified PBFT, modified RIPPLE and modified DPOS algorithms. Hence, we want to describe DDPOS algorithm first, and then describe the four modified algorithms it contains. Besides, figure 13, 14 and 15 were removed and their information were summarized in Table 2.
Question 5:
The reviewer’s comment: A separate “Discussion” section is recommended.
The authors’ Answer: Thanks for your kind suggestion. Now the “Discussion” section was supplemented, which can be seen in section 5.
Question 6:
The reviewer’s comment: Conclusion section is consistent with the results obtained.
The authors’ Answer: Thanks for your kind suggestion. Now we have supplemented and modified relevant descriptions, which can be seen in section 6.
Thank you for your suggestions. All your suggestions are very important, which have important guiding significance for my thesis writing and scientific research!
Round 2
Reviewer 1 Report
Suggestions:
1)At Discussion. ....These five algorithms are analyzed from four three aspects: code quantity, resource occupancy and security.
2) At the end of the conclusions. I suggest "The results show that the overall performance of DDPOS is very high compared to the others presented here (or the best of those presented here)."
Because "optimal" means that there is no other algorithm better than this one. And that needs a mathematical proof.
Author Response
Reply to reviewer 1
Dear reviewer:
I am very grateful to your comments for the manuscript. According with your advice, we amended the relevant part in manuscript. All your questions were answered below.
Question 1:
The reviewer’s comment: At Discussion. ....These five algorithms are analyzed from four three aspects: code quantity, resource occupancy and security.
The authors’ Answer: Thanks for your kind suggestion. We made mistakes in the process of writing. Now we have changed the word “four” into “three”.
Question 2:
The reviewer’s comment: At the end of the conclusions. I suggest "The results show that the overall performance of DDPOS is very high compared to the others presented here (or the best of those presented here)." Because "optimal" means that there is no other algorithm better than this one. And that needs a mathematical proof.
The authors’ Answer: Thank you for your kind suggestion. We used the wrong word “optimal”. Your proposal will be very precise. Now, according to your suggestion, we have modified the relevant sentence at the end of the “Conclusion” section.
Thank you for your suggestions. All your suggestions are very important, which have important guiding significance for my thesis writing and scientific research!